# A behavioral signature for quantifying the social value of interpersonal relationships with specific others
João F. Guassi Moreira ✉ & Carolyn Parkinson

The idea that individuals ascribe value to social phenomena, broadly construed, is well-established. Despite the ubiquity of this concept, defining social value in the context of interpersonal relationships remains elusive. This is notable because while prominent theories of human social behavior acknowledge the role of value-based processes, they mostly emphasize the value of individual *actions* an agent may choose to take in a given environment. Comparatively little is known about how humans value their interpersonal relationships. To address this, we devised a method for engineering a behavioral signature of social value in several independent samples (total $N = 1111$). Incorporating the concept of opportunity cost from economics and data-driven quantitative methods, we derived this signature by sourcing and weighting a range of social behaviors based on how likely individuals are to prioritize them in the face of limited resources. We examined how strongly the signature was expressed in self-reported social behaviors with specific relationship partners (a parent, close friend, and acquaintance). Social value scores track with other aspects of these relationships (e.g., relationship quality, aversion to losing relationship partners), are predictive of decision preferences on a range of tasks, and display good psychometric properties. These results provide greater mechanistic specificity in delineating human value-based behavior in social contexts and help parse the motivational relevance of the different facets that comprise interpersonal relationships.

Humans are an intensely social species, so much so that lacking social connection and strong relationships is considered a risk factor for negative mental and physical health outcomes[1,2]. This has led to the declaration of social connection a basic human need[3,4]. It is thus unsurprising that social stimuli and interactions appear to be intrinsically rewarding. Evidence from human neuroimaging studies suggests various types of social stimuli (e.g., interactions, information, feedback, images of high-status individuals) elicit strong responses in reward-related brain regions[5–7], and studies using behavioral economic paradigms have revealed that people are often willing to forego money or other rewards in favor of social stimuli[8–10].

Despite the fact that social connection is a human need, the amount and quality of interaction desired, and the manner in which it occurs, varies across individuals[11–15]. Individuals differ in how much they approach social settings[16,17], how they tolerate different types of social interactions[11,12] (e.g., with familiar others vs with strangers), how they orient themselves towards outcomes impacting other people[18–20], and so on. Observations of between- and within-individual variability in the desire for social connection has sparked the idea that individuals vary in the degree to which they value social phenomena, and has subsequently catalyzed work attempting to formally

quantify such value. Scientists from a number of disciplines within the behavioral sciences (e.g., psychology, economics, sociology) have developed methods for quantifying the value of social phenomena, such as information sharing, decisions, and interactions.

By comparison, relatively little work has tried to quantify the social value of *social relationships*. Prior work has attempted to formally quantify the value of individual social interactions, but has not focused on calculating the value of specific, individual interpersonal relationships[21–23]. While models stemming from this work typically include a term to model any residual influence of the social 'context' or 'background' on the value of individual social interactions, these terms do not isolate the value of individual interpersonal relationships. These aspects of past work effectively assume the social value one places on specific relationships is constant across relationship partners or is indistinguishable from other contextual factors. Other work has focused on quantifying the value that one places on another's welfare by estimating the amount of money one would forgo for themselves to earn money for another individual[24–26]. Yet parameters from these models are sensitive to transient affective changes and socioeconomic background,

Department of Psychology, University of California, Los Angeles, CA, 90095, USA. ✉e-mail: jguassimoreira@ucla.edu

indicating they are not reliable or generalizable measures of the value one places on an interpersonal relationship.

In the current study, we developed and validated a procedure for deriving a behavioral signature of interpersonal value that facilitates quantifying the value of one's interpersonal relationships. Humans expend finite resources, such as time and energy, when cultivating and maintaining their interpersonal relationships[27]. Therefore, we chose to ground our approach to defining and measuring social value in the concepts of opportunity cost and scarcity. This conceptually mirrors how value has been fruitfully operationalized in related domains[28–30]. Specifically, we defined social value in terms of how individuals were willing to allocate their time and effort when engaging with others. However, we viewed it as insufficient to simply define value in terms of how often individuals spent time or energy engaging with others—individuals may frequently engage with others for incidental reasons or out of necessity. Thus, such an approach would fail to capture how much an individual spends time doing valued or desired activities with a social partner.

Instead, we devised a method for engineering a *behavioral signature* of social value. This signature is a collection of common social activities and associated weights that quantify the activities' value to individuals in the face of finite time and effort, and represents an idealized pattern of time allocation to social activities. We used this pattern to quantify the value of individual social partners by examining how much this behavioral signature was expressed in individuals' self-reported behaviors with those partners. The entire process of engineering the behavioral signature involved *sourcing activities* that would comprise the signature, *deriving weights* that quantified the extent to which individuals are willing to prioritize engagement in a particular activity in the face of opportunity cost, collecting data on *how likely* individuals are to complete each activity with specific social partners, and *computing the expression* of the behavioral signature in the aforementioned likelihoods. Each weight that comprises the signature reflects the likelihood of prioritizing a given activity relative to all others; collectively they reflect a hierarchy of preferences for each activity. This behavioral signature of social value is intended to represent an idealized allocation of finite leisure time across a variety of possible activities. We argue that the extent to which one's activities with a given social partner conforms to this pattern should be reflective, to a meaningful degree, of the value one places on said partner (albeit not perfectly so). We intentionally chose to ground the behavioral signature in how people spend their free time (e.g., engaging in activities), which is typically scarce, as opposed to resource sharing, to avoid conflating individual differences in resource availability with individual differences in social value. Relatedly, we emphasize that the method we present and validate here is meant to capture value based on *behavior*. We deliberately avoided asking people to explicitly estimate the value of their relationships given that introspective awareness is often limited[31] and that repeatedly asking people about their attitudes concerning a relationship can change such attitudes[32].

Here we develop and refine this procedure for quantifying a behavioral signature of interpersonal social value using three categories of social partners: parents, friends, and acquaintances. We then validate the subsequent social value scores in ten subsamples across two phases (exploratory, confirmatory) by correlating the scores with various measures of social behaviors and attitudes towards a parent, a friend, and an acquaintance nominated by participants.

## Methods

*Overview.* The present study has two phases: exploratory and confirmatory. The exploratory phase established the procedure for quantifying a behavioral signature of social value and included preliminary validation (correlating social value scores with relevant metrics of social behavior). The confirmatory phase involved replicating the preliminary analyses from the prior phase in a larger sample and introducing additional metrics to further validate the social value scores. The full study contains data from $N = 1,111$ participants ($n_{exploratory} = 476$, $n_{confirmatory} = 635$). Social value scores were computed for three categories of familiar others: parents,

friends, and acquaintances (every participant had to nominate one of each). Demographic information for all samples is included in Table 1. All study data, code, and study materials are publicly available on the Open Science Framework (OSF; osf.io/rjqpc). We pre-registered the confirmatory phase of the study on August 3rd, 2022 (see OSF); deviations from the pre-registration are listed in the Supplement. The method we describe here is meant to yield one social value score for each specific relationship partner (three, in this case). We use the terms 'relationship partners', 'social partners' and 'familiar others' interchangeably where appropriate. All practices and procedures described herein were approved by the UCLA Institutional Review Board (IRB#21-002041). All participants provided informed consent in accordance to procedures approved by the IRB. Participants were either compensated with course credit or in USD ($) based on guidelines offered by the online data collection platform (MTurk or Prolific).

*Exploratory Phase*
The exploratory phase entailed three procedures: (i) sourcing activities, (ii) deriving value weights for said activities with which to subsequently compute social value scores, and (iii) running statistical tests to establish the validity of the social value scores. A schematic of the process is depicted in Fig. 1.

*Sourcing Activities.* For the activity sourcing procedure, participants were instructed to list up to 25 activities that they have completed in the past with another person. Participants were told they could be as broad or narrow (e.g., "watching television" vs "watching *The Sopranos*") and as redundant (e.g., listing both "attending a hip-hop concert" and "attending a jazz concert") as they wished. Members of the research team then pared the entire pool of activities into higher-level categories (e.g., "browsing Instagram" and "browsing TikTok" became "browsing social media"). Describing activities at a slightly higher level of abstraction facilitated data aggregation across respondents and their social partners, and in so doing, created a behavioral signature of social value that could generalize across social partners and respondents. We reasoned that activity descriptions that were too granular would limit the aggregation of data across people, and that activity descriptions that were too broad would lead to the loss of meaningful distinctions between activities. To evaluate the robustness of the method, we sourced two different sets of activities from different samples (UCLA's online departmental subject pool, referred to as SONA, 70 items; Amazon's Mechanical Turk, referred to as MTurk, 56 items). Demographic data and self-reported Big Five personality traits were collected to characterize the samples (Table 1). The MTurk set of activities is presented in Table 2, the SONA activity set is shown in Supplementary Table 1.

Activities were collected on the Qualtrics survey platform from two separate samples—one from SONA and another from MTurk. The prompt for sourcing activities follows: *"For this study, we want to learn more about how individuals spend their time with other people. In the text boxes below, please list activities you have done in the past with another person. You are welcome to be as specific (e.g., 'watching Love Is Blind') or as broad (e.g., 'watching television') as you wish. Please do your best to list 25 activities, even if some are a little redundant (e.g., 'attending a jazz concert' and 'attending a hip-hop concert'). You may list activities that intrinsically involve other people (e.g., 'asking someone for advice about a problem you are facing') or activities that can be done with other people or alone (e.g., 'swimming at the beach'). It is OK to list activities that might not be considered fun or recreation (e.g., 'replacing my car's windshield wipers'). Finally, you are welcome to list activities that you do at whatever frequency (e.g., regularly v seldom)."*

*Deriving Activity Weights and Computing Social Value Scores.* Deriving Activity Value Weights Using Maximum Difference Scaling. We used maximum difference (MaxDiff) scaling (also known as Best-Worst Scaling[33];) to derive value weights for each of the activities to be included in our behavioral signature of social value. The following section describes the background of MaxDiff and our rationale for using it, the study design that we employed throughout the current investigation, and details of analyses of MaxDiff data using a hierarchical Bayesian approach.

**Table 1 | Sample demographics and characteristics**

| Sample | | | | |
|---|---|---|---|---|
| | **Activity generation (AG)** | **MaxDiff (MD)** | **Exploratory validation (EV)** | **Confirmatory validation (CV)** |
| Sample Size | $N = 79$ | $N = 128$ | $N = 269$ | $N = 635$ |
| Source | SONA, MTurk | SONA, MTurk | SONA, MTurk | MTurk, Prolific |
| Sex | F = 54.43%<br>M = 45.57% | F = 48.44%<br>M = 50.70% | F = 59.11%<br>M = 40.89% | F = 45.19%<br>M = 54.81% |
| Average Age (SD) | 26.87 (7.50) | 33.33 (11.60) | 23.46 (3.33) | 25.33 (3.01) |
| Race | Af Am – 2.5%<br>Asn – 19.0%<br>NH/PI – 0.0%<br>W – 62.0%<br>NA/AN – 2.5%<br>O – 7.6%<br>M – 6.3% | Af Am – 7.8%<br>Asn – 16.4%<br>NH/PI – 0.0%<br>W – 68.0%<br>NA/AN – 0.0%<br>O – 1.5%<br>M – 6.3% | Af Am – 8.2%<br>Asn – 24.9%<br>NH/PI – 0.7%<br>W – 62.8%<br>NA/AN – 1.5%<br>O – 4.1%<br>M – 4.8% | Af Am – 13.91%<br>Asn – 11.27%<br>NH/PI – 0.0%<br>W – 65.3%<br>NA/AN –1.5%<br>O – 2.0%<br>M – 5.3% |
| Ethnicity (Hispanic/Latinx) | 17.7% | 4.7% | 19.0% | 14.35% |
| Big Five<br>Mean (SD) | Ext: 3.15 (0.96)<br>Agr: 3.96 (0.67)<br>Cts: 3.98 (0.77)<br>Nrt: 2.67 (1.03)<br>Opn: 3.82 (0.69) | Ext: 2.95 (1.02)<br>Agr: 3.91 (0.72)<br>Cts: 4.00 (0.78)<br>Nrt: 2.68 (1.04)<br>Opn: 3.54 (0.73) | NA | NA |
| Loneliness<br>Mean (SD) | NA | 2.15 (0.62) | NA | NA |

*Note.* SONA refers to the UCLA undergraduate psychology subject pool; MTurk refers to Amazon's Mechanical Turk research crowdsourcing platform; Prolific refers to the research crowdsourcing platform. Age is reported in years. 'Ext' refers to extraversion/introversion; 'Agr' refers to agreeableness/antagonism; 'Cts' refers to conscientiousness/directionlessness; 'Nrt' refers to neuroticism/emotional stability; 'Opn' refers to openness/close-mindedness. 'SD' refers to standard deviation. 'Af Am' refers to African American; 'Asn' refers to Asian; 'NH/PI' refers to Native Hawaiin/Pacific Islander; 'NA/AN' refers to Native American/Alaskan Native; 'W' refers to White; 'O' refers to other; 'M' refers to mixed race. Racial percentages that do not add up to 100% reflect the presence of participants who declined to respond. Sample sizes reflect the number of participants recruited during the data collection period for the listed study phase. Sample sizes may vary between analyses depending on data quality. The CV sample size excludes fraudulent responses from MTurk. Sex was self-reported by participants; any percentages for sex that do not add up to 100% reflect participants who declined to report their sex.

*Background and Rationale.* Our strategy for defining a behavioral signature of social value involved creating a list of activities that varied in the degree of value that participants ascribed to them (defined in terms of how likely they were to prioritize each activity given limited time), measuring how often participants engaged in those activities with each of their social partners, and then using this information to generate an estimate of the social value of each of their social partners.

We used a MaxDiff design to generate estimates of the relative values of the various activities. MaxDiff designs, originating in marketing and applied economics, ask respondents to select the most appealing and least appealing items in a set based on a specific feature of interest. This design forces respondents to reveal trade-offs between items and aligns with the economic definition of value rooted in the concept of opportunity cost. In contrast, other commonly used approaches (e.g., Likert scales) are prone to scale bias[34,35] and cannot easily incorporate trade-offs between activities or opportunity cost. MaxDiff designs have been widely used in marketing, applied economics, and other disciplines[33,36,37]. This rationale is explained in greater detail in the Supplement.

*Maximum Difference Scaling Study Design.* We presented participants with sets of activities along with the following prompt. "*Suppose you had a couple hours of time wherein you had no obligations or commitments. Which of the following activities are you most likely to do in this time? Which would you be least likely to do?*" Each set was comprised of four activities selected at random from a broader pool. Activities were presented vertically in the middle of the screen flanked by a column of response options on either side (*least likely* on the left, *most likely* on the right). Participants could only select one activity per column, resulting in two choices for each set. Participants were presented with 53 or 42 sets, depending on the activity pool being rated (see Supplement for details).

We administered this MaxDiff design twice, once for each pool of activities. The first administration was on a sample of UCLA undergraduates through the psychology department's online subject pool (SONA). Twenty-eight (23 female, 5 male, $M_{age} = 20.64$ years) students were remotely administered a Qualtrics survey containing the MaxDiff

design with activities generated from a separate sample of undergraduate psychology students described in the "Sourcing Activities" section. The second administration was on a sample of 50 MTurk Workers (17 female, 33 male, $M_{age}$: 36.57 years) using activities generated by a separate sample of MTurk workers described in the "Sourcing Activities" section. Demographic data, self-reported Big Five personality traits, and self-reported loneliness were collected to characterize the samples (Table 1; see Supplement for information on the measures). An additional 50 MTurk workers completed a Max Diff study from the SONA-Sourced Activities. However, we had difficulty successfully fitting the model to this dataset due to the high number of parameters and abandoned the endeavor.

*Hierarchical Analysis of Maximum Difference Scaling Data.* Activity weights for each item were obtained using hierarchical Bayesian logistic regression. The resultant coefficients (visualized in Supplementary Fig. 1) from this model represent the mean likelihood of selecting an activity as most likely (or least likely, if the sign is inverted). The magnitude of the coefficient codes for the likelihood; the sign codes for the direction (positive maps onto 'most likely to do', negative corresponds to 'least likely to do'). The zero-sum nature of the maximum difference task means that activity weights can be ordinally ranked to reveal the highest-ranking activities. These coefficients are the 'activity weights' that comprise our behavioral signature, indicating the relative preference for each activity (scaled in log odds). In depth modeling details follow.

Activity weights for each item were obtained using hierarchical Bayesian logistic regression with a set of overparameterized dummy codes. The data were structured in a particular manner for this analysis. The dependent variable was comprised of a column that represented the response to an activity within a given set presented to a given participant. Responses were binary-coded such that a '1' indicated that the participant endorsed the activity as least *or* most likely, and a '0' signified the participant did not make such an endorsement. The design matrix was created with columns representing overparameterized dummy code indicators for each activity. The dummy code (-1, 0, 1) indicated whether an activity appeared in a set. Each activity contributed two rows in the matrix, one for the 'most

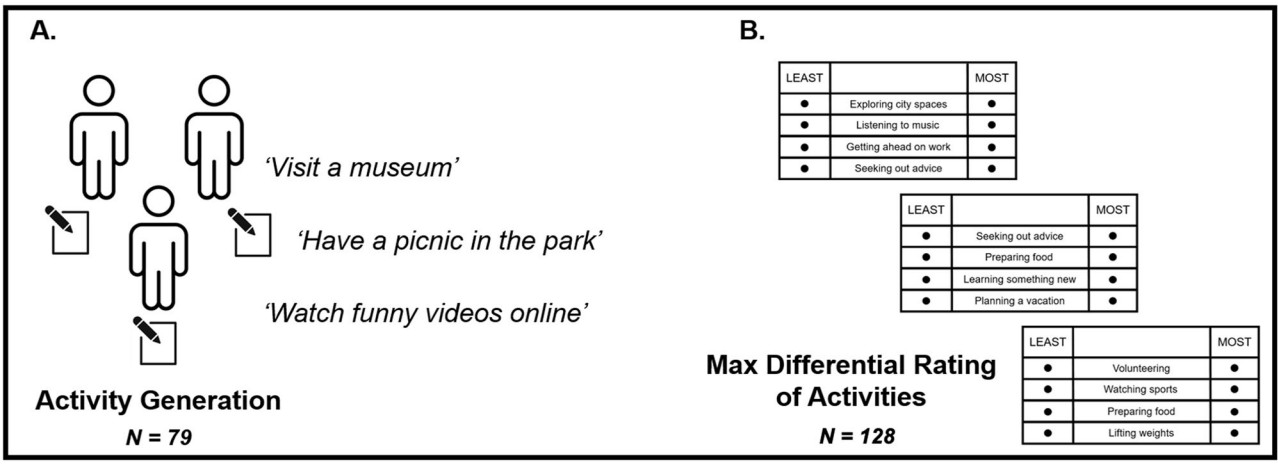

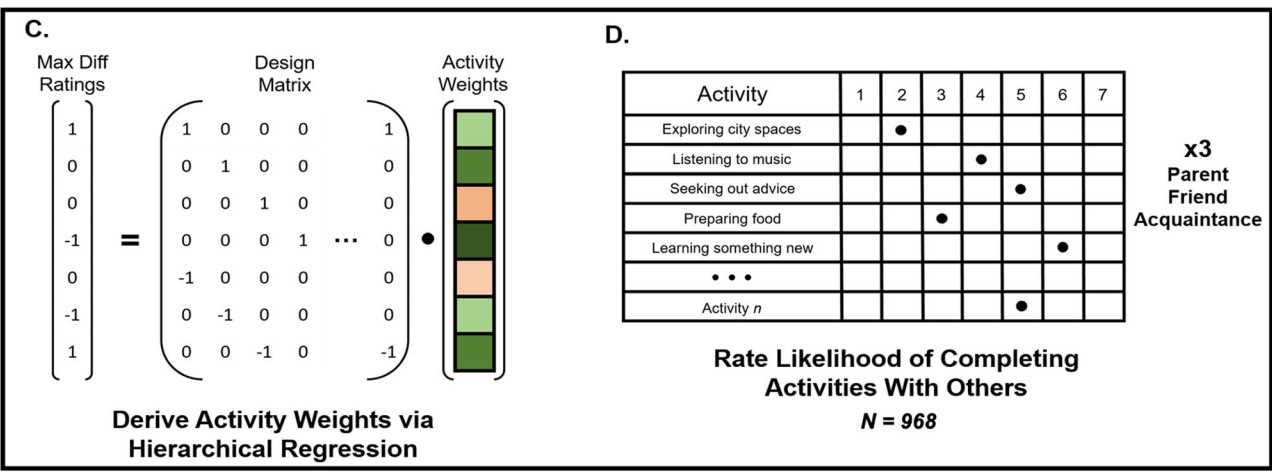

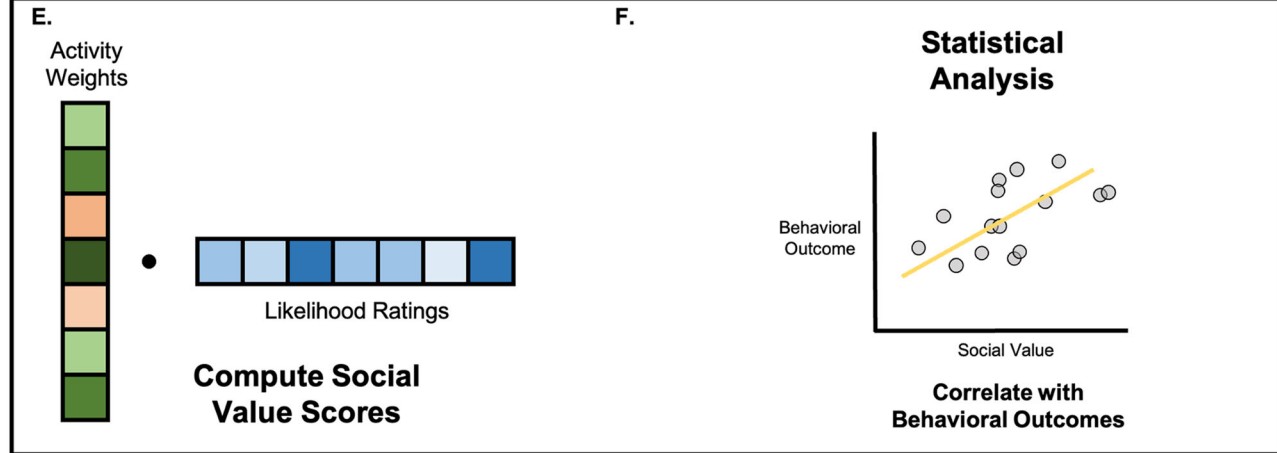

**Fig. 1 | Overview: quantifying a behavioral signature of social value.**
**A** Participants were asked to list potential activities that they could complete with other individuals in whatever level of detail they wish. Members of the research team consolidated activities into broader categories (e.g., 'Watching *The Sopranos*' or 'Watching *Love is Blind*' become 'Watching television'). **B** An independent set of participants completed best-worst ratings of activities in a traditional maximum difference design that allows for (**C**) derivation of activity weights that produces a behavioral signature of social value. The signature is applied by having novel participants rate how likely they are to complete each activity in the signature with a given social partner, such as a parent or friend (**D**), then taking the dot product of these likelihood ratings and activity weights (**E**), yielding a scalar value quantifying the social value that a given participant ascribes to their relationship with a given a social partner. Social value scores were validated in series of statistical analyses involving behaviors and attitudes towards others (**F**) across exploratory and confirmatory samples. For robustness, two separate sets of activities were sourced and weighted.

likely' response and one for the 'least likely' response in the MaxDiff design. In the matrix, a '1' was entered for the 'most likely' option, and a '−1' for the 'least likely' option. Other entries in the row were set to zero. Supplementary Table 2 provides a simplified example of the response variable and design matrix. The design matrix had either 56 or 70 dummy codes, depending on

whether SONA (UCLA undergraduate psychology subject pool) or MTurk (Amazon's Mechanical Turk) data were analyzed. An example of a model design matrix is included in the Supplement (Supplementary Table 2).

Notably, the model is estimated without an intercept, given the set of overparameterized dummy codes (the number of dummy codes is equal to

## Table 2 | MTurk-generated activities and MTurk-rated weights

| Activity | Raw Weight | Subject Variance | Scaled Weight |
|---|---|---|---|
| Attending a musical event | -0.29 | 0.76 | -0.38 |
| Arts and crafts (ex: knitting, painting) | 0.09 | 1.06 | 0.08 |
| Attending a show/event (ex: theater, musical performance) | -0.08 | 0.73 | -0.11 |
| Attending a social gathering | -0.28 | 0.68 | -0.41 |
| Attending a sporting event | -0.75 | 1.36 | -0.55 |
| Attending an academic event (ex: a public lecture) | -0.53 | 0.42 | -1.26 |
| Attending to one's health (ex: scheduling appointments, visiting a medical professional) | -0.33 | 0.57 | -0.58 |
| Browsing the internet | 1.62 | 0.63 | 2.57 |
| Building or repairing something (ex: car, fence, treehouse) | -0.06 | 1 | -0.06 |
| Celebrating a recent event (ex: birthday, accepting a new job) | -0.32 | 0.21 | -1.52 |
| Cleaning | 0.35 | 1.15 | 0.30 |
| Communicating virtually with someone (ex: zoom, facetime) | -0.31 | 0.75 | -0.41 |
| Completing personal chores/tasks | 0.66 | 1.13 | 0.58 |
| Conversating with someone | 0.45 | 0.26 | 1.73 |
| Dancing | -1.01 | 0.57 | -1.77 |
| Discussing personal matters | 0.02 | 0.2 | 0.10 |
| Engaging in politics (ex: volunteering for a campaign, reading political articles) | -0.99 | 1.5 | -0.66 |
| Going for a bike ride | -0.18 | 0.99 | -0.18 |
| Going for a drive | 0.37 | 0.59 | 0.63 |
| Going for a run | -0.31 | 0.89 | -0.35 |
| Going for a walk | 0.85 | 0.48 | 1.77 |
| Going out for a drink | -0.78 | 1.22 | -0.64 |
| Going out to eat | 0.55 | 0.87 | 0.63 |
| Going to a cafe | -0.01 | 0.44 | -0.02 |
| Hosting a social gathering | -0.87 | 0.35 | -2.49 |
| Lifting weights | -0.58 | 1.44 | -0.40 |
| Listening to a podcast | 0.31 | 0.74 | 0.42 |
| Listening to music | 1.06 | 0.89 | 1.19 |
| Participating in a religious activity | -1.75 | 1.75 | -1.00 |
| Participating in online games/activities | 0.7 | 1.31 | 0.53 |
| Participating in recreational games (ex: bowling, chess, mini-golf) | 0.06 | 0.55 | 0.11 |
| Playing a sport | -0.55 | 1.04 | -0.53 |
| Playing with a pet(s) | 0.63 | 1.2 | 0.53 |
| Preparing food (ex: cooking, baking) | 0.8 | 0.98 | 0.82 |
| Reading | 0.62 | 1.14 | 0.54 |
| Receiving a cosmetic treatment (ex: hair styling) from a professional | -0.68 | 0.92 | -0.74 |
| Resting, sleeping, or relaxing | 0.9 | 0.42 | 2.14 |
| Seeking out or receiving advice | -0.26 | 0.18 | -1.44 |

## Table 2 (continued) | MTurk-generated activities and MTurk-rated weights

| Activity | Raw Weight | Subject Variance | Scaled Weight |
|---|---|---|---|
| Self-applying a cosmetic treatment (ex: painting nails, styling hair) | -0.62 | 1.26 | -0.49 |
| Shopping for leisure | 0.42 | 0.93 | 0.45 |
| Shopping for necessities (ex: groceries) | 0.59 | 0.75 | 0.79 |
| Singing | -0.76 | 0.8 | -0.95 |
| Smoking tobacco or marijuana recreationally | -1.41 | 1.75 | -0.81 |
| Spending time at an amusement park | -0.3 | 0.7 | -0.43 |
| Spending time in public spaces (ex: park, city square, zoo) | 0.13 | 0.36 | 0.36 |
| Spending time on a hobby | 1.27 | 1.77 | 0.72 |
| Spending time on artistic activities | 0.22 | 0.63 | 0.35 |
| Studying | -0.36 | 0.52 | -0.69 |
| Taking care of a pet(s) | 0.67 | 1.07 | 0.63 |
| Traveling/vacationing | -0.12 | 0.74 | -0.16 |
| Volunteering | -0.77 | 0.23 | -3.35 |
| Watching a movie in a theater | -0.11 | 1.44 | -0.08 |
| Watching something on a streaming service | 0.97 | 0.81 | 1.20 |
| Watching sports | -0.13 | 1.72 | -0.08 |
| Watching television | 0.95 | 1.15 | 0.83 |
| Working | 0.33 | 0.87 | 0.38 |

*Note*. MTurk refers to Amazon's Mechanical Turk research crowdsourcing platform. MTurk participants generated activities, which experimenters pared down into slightly more abstract descriptions, as described in the main text, to facilitate consolidation of activities across respondents. A separate sample of MTurk participants rated the activities in a MaxDiff design to yield the data that were used to calculate the weights that were used to compute social value scores.

the number of activities). This allows the coefficients to represent the mean likelihood of selecting an activity as 'most likely' or 'least likely', depending on its code. These coefficients form the 'activity weights' in our behavioral signature, indicating the relative preference for each activity (scaled in log odds). Coefficients were allowed to vary randomly across participants. Posterior means of the 'fixed effects' were scaled by the posterior variances of the 'random effects' to account for individual differences in preferences among participants. Various sensitivity analyses revealed that this scaling and inclusion of activities with high random effect variance did not appreciably change the rank order among participants for any of the social targets tested here. More specifically, correlations between social value scores computed using our original approach (with all activities included and with scaled weights) and social value scores computed without scaling weights by their variance and/or by excluding activities with high-variance weights ranged from $r = 0.81$ 0.98 across social partners. We implemented this model on the two datasets described above. We used the *brms* package to estimate the model (8 chains (1 per core), 1,000 iterations/chain, 500 warmup iterations, no thinning, target average acceptance proposal probability of .95, a step-size of .05, max tree depth of 15). Data were modeled as being drawn from a Bernoulli distribution, with standard normal priors specified for activity coefficients. The hierarchical nature of the model and use of weakly informative priors help prevent overfitting.

*Computing Social Value Scores.* Our process of computing social value scores involves determining how much the behavioral signature of value is expressed in an individual's behaviors involving a specific social partner. We drew inspiration from pattern expression analyses in the human cognitive neuroscience literature[38,39]. In cognitive neuroscience, pattern expression

**Table 3 | Paired differences in social value scores and relationship quality scores (exploratory sample)**

| Comparison (Subsample) | Social value | Relationship quality |
|---|---|---|
| Parent–Friend (Sona-1) | -0.43 [-0.73, -0.15] | -0.70 [-1.04, -0.38] |
| Parent–Acquaintance (Sona-1) | 0.48 [0.18, 0.77] | 0.67 [0.35, 0.98] |
| Friend–Acquaintance (Sona-1) | 0.70 [0.38, 1.03] | 1.39 [0.94, 1.80] |
| Parent–Friend (Sona-2) | -0.17 [-0.47, 0.15] | -0.62 [-1.00, -0.26] |
| Parent–Acquaintance (Sona-2) | 0.68 [0.30, 1.03] | 0.22 [-0.10, 0.52] |
| Friend–Acquaintance (Sona-2) | 0.73 [0.36, 1.10] | 1.36 [0.93, 1.84] |
| Parent–Friend (Sona-3) | -0.14 [-0.31, 0.02] | -0.78 [-0.98, -0.57] |
| Parent–Acquaintance (Sona-3) | 0.83 [0.61, 1.05] | 0.64 [0.43, 0.84] |
| Friend–Acquaintance (Sona-3) | 1.06 [0.84, 1.32] | 1.49 [1.22, 1.74] |
| Parent–Friend (MTurk-1) | 0.08 [-0.11, 0.25] | -0.51 [-0.70, -0.32] |
| Parent–Acquaintance (MTurk-1) | 0.42 [0.21, 0.60] | 0.18 [0.01, 0.37] |
| Friend–Acquaintance (MTurk-1) | 0.37 [0.17, 0.55] | 0.85 [0.64, 1.07] |
| Parent–Friend (MTurk-2) | 0.01 [-0.18, 0.21] | -0.48 [-0.68, -0.28] |
| Parent–Acquaintance (MTurk-2) | 0.41 [0.20, 0.63] | 0.07 [-0.12, 0.28] |
| Friend–Acquaintance (MTurk-2) | 0.36 [0.16, 0.55] | 0.76 [0.55, 0.99] |

*Note.* Brackets represent 89% HDIs of posterior probability distributions. Subsamples are listed in parentheses; 'Sona' or 'MTurk' reflect where the sample completing the likelihood ratings was recruited from. Sona-2 was administered MTurk-sourced activities; MTurk-2 was administered Sona-sourced activities. All other samples were administered activities sourced from a different sample within the same population. 'Relationship Quality' refers to mean scores from the IPPA.

analyses answer the question of how much a particular psychological process is expressed in a map of neural activity by examining the similarity between the map of neural activity and a neural signature (a model of how the brain encodes a psychological state in a multivariate pattern). Pattern expression scores are usually computed as the dot product between the values of brain activity and model weights for each voxel in the brain.

We applied the same logic when computing social value scores: the social value of a given social partner is determined by computing how much a behavioral signature of value is 'expressed' in the behaviors that one engages in with a social partner. To do so, we asked participants to indicate how likely they would be to complete the activities in the behavioral signature with a given social partner, assuming 'average' conditions and relatively easy access to the partner. We term these data 'likelihood ratings'. Likelihood rating surveys were administered with the following prompt. *"Assume that you had a free day with no imminent obligations or commitments. Assume that this day is taking place during an 'average' month for you. How likely are you to engage in the following activities with the [PERSON] you nominated? In this scenario, assume that you live near your [PERSON] and that you could see them relatively easily if you wanted to."* [PERSON] was replaced by 'parent', 'friend' or 'acquaintance'.

Participants completed likelihood ratings 3 times, once for each of the three social partners examined here (parent, friend, acquaintance). Social value scores are computed by taking the dot product between the likelihood ratings and activity weights comprising the behavioral signature of social value. That is, there are three sets of likelihood ratings per participant (one for their parent, one for their friend, and one for their acquaintance) and the same behavioral signature is applied to all activity ratings for all participants, producing three social value scores for each participant.

*Validating Social Value Scores.*

We examined correlations between social value scores and other ways of evaluating interpersonal relationships to validate our novel method of calculating social value. Results from the exploratory sample reported here consisted of 5 independent subsamples that were either collected from an online subject pool (SONA) or MTurk. One of two sets of activity weights, sourced from different populations were used (SONA or MTurk). Analyses were run separately by sample. Aggregate statistics are reported here for

clarity; disaggregated statistics are reported in the Supplement. Sample sizes for each sample, in order of collection, were $N = \{30, 24, 76, 75, 64\}$.

Participants completed several additional measures that aided in validation of social value scores: Self-reported relationship quality with each social partner, questions about how much time they actually spent and would ideally spend with each social partner, social loss aversion for each social partner (i.e., how upset one would be if they could no longer spend time with an individual), one-shot dictator games where participants made decisions about how to allocate hypothetical monetary resources among all possible pairs of social partners, and a forced choice question about whom they would rather spend time with (involving all possible pairings). Details about these each of these measures are provided in the Supplement.

We first checked if social value scores were correlated with relationship quality, social loss aversion, and how much time (ideal and actual) individuals spent with each social partner. Theoretically, social value ought to be correlated with these measures to some degree. These analyses were repeated in 3 sets, one for each social partner, using robust Pearson correlations estimated with weakly informative *t*-distributed priors (see code; osf.io/q2npw). Then, we ran multiple regression analyses to determine whether social value scores could predict social decision behavior involving monetary (dictator game) and social (forced choices regarding spending time) outcomes among pairs of social partners. Similarly, individual differences in the social value that one ascribes to specific relationship partners should theoretically track with individual differences in social decision preferences between said partners. Weakly informative priors ($N(0,1)$) were placed on regression coefficients. Social value scores and dictator game allocations were standardized prior to analysis. Logistic regression was used for the forced choice question about whom one would rather spend time with. Additionally, while not a primary focus in validating our approach, we also computed pairwise differences between the scores for the three types of social partners to examine whether participants generally valued one type of social partner more than another. These analyses are meant to serve as a sanity check to confirm distinctions among different relationship categories (e.g., close vs distant relationships). These results are listed in Table 3.

*Inferential Criteria.* We summarized posterior distributions using the mean of the relevant statistic (e.g., Cohen's *d*, regression coefficient) and an 89% highest density credible interval (HDI). We used 89% credible intervals upon the recommendation that wider intervals (e.g., 95%) are more sensitive to Monte Carlo sampling error[40,41]. We used two criteria to perform inferences: (i) examining whether the HDI contained zero, and (ii) Kruschke's Region of Practical Equivalence Method (ROPE)[42,43]. ROPE was defined as the range between −0.1 and 0.1 for all analyses. Evidence was judged to be robust if the HDI did not include zero or the HDI fell outside of ROPE; evidence was judged to be moderate if part of the HDI fell outside of ROPE[44]. While credible intervals are often interpreted similarly to confidence intervals, we note that a credible interval overlapping with zero is not necessarily indicative of a null statistical effect. Credible intervals simply help summarize the distribution of evidence over parameter space, and overlap with zero does not rule out evidence in favor of a meaningful effect.

**Confirmatory phase**

We collected additional data to confirm and expand upon the results observed in the exploratory phase of the study. Given that some statistical associations observed in the previous phase were in the expected direction but had large posterior variances, we decided to drastically increase the sample size ($N = 635$ across four independent subsamples). Participants were recruited from MTurk and Prolific.

Methods for the confirmatory phase were mostly consistent with the validation step of the exploratory phase. One key difference was the recruitment of much larger sample sizes on a different data collection platform. A second key difference is that we added a multi-trial social decision-making task and novel measures of affiliative social behaviors to expand upon the results in the exploratory phase (see Supplement for further details). To reduce task demands after adding these novel items,

questions regarding time spent with each partner were dropped. These aspects of the confirmatory phase are reviewed below.

*Data Collection Platform.* We initially began this data collection on MTurk. However, it quickly became apparent that our responses for this particular instance of data collection were marred by a high rate of fraud (indicated by written response to open-ended questions). Specifically, we found evidence to suggest that a single user, or a group of users, were repeatedly signing up for our study under multiple accounts. Although we cleaned existing data and began screening subsequent responses according to recommendations set forth by others who experienced the same problem[45,46], the rate of fraudulent responses remained quite high and considerably impeded the study's progress. We thus abandoned data collection on the MTurk platform and switched to using the Prolific platform, where we did not encounter the same issue.

We decided to analyze the MTurk data, according to our pre-registration plan, in addition to the Prolific data. We kept the screened and cleaned MTurk data so that these data could serve as an additional test of generalizability between online populations. Thus, the results reported in this confirmatory phase come from four subsamples, fully crossed between the participant pools from which participants were drawn to obtain lists of activities (Prolific, MTurk) and to validate social value scores (Prolific, MTurk). Notably, although concerns about fraudulent responses led to a final MTurk sample for validating social value scores that was smaller than planned ($N = 181$; $N = 82$ with SONA-sourced weights, $N = 99$ with MTurk-sourced weights), by conducting additional recruitment via Prolific, we obtained a final sample of $N = 454$ ($N = 233$ with SONA sourced weights, $N = 221$ with MTurk sourced weights), consistent with our pre-registered data collection plan.

*Multi-Trial Social Decision-Making Paradigm.* We expanded the scope of our investigation by examining if and how social value scores tracked with social decision preferences in the context of multi-trial tasks. Multi-trial tasks are a useful complement to one-shot tasks because they accommodate intra-individual variability in decision behavior, are more internally consistent given the volume of trials, and allow researchers to vary decision-level features (e.g., reward) which helps estimate choice preferences that are generalizable over various contexts. Participants completed several runs of an existing multi-trial social decision-making paradigm[47]. Similar to our one-shot questions, the paradigm presented participants with conflicting options for allocating money or time between all three possible pairings of social partners (one pair per trial). However, unlike the one-shot questions, in this paradigm, there were multiple trials per pair, each varying in the amount of money to be allocated and the timing of the allocations. Allocations in this paradigm are made in a delay discounting format, allowing us to examine how the relationship between social value and social choice preference generalizes to a different type of decision-making. Greater details on this paradigm and its statistical modeling details are included in the Supplement.

*Preferences for Social Affiliative Behaviors.* We also expanded the scope of our investigation by collecting data on additional forced-choice questions about participants' preferences regarding a variety of social affiliative behaviors. Similar to the questions where participants chose between two potential social partners to spend time with, these questions asked participants which of two social partners they would favor for a given social affiliative behavior. Six such behaviors were examined here—advice seeking, celebrating something, sharing positive news, sharing negative news, lending money, and having dinner. More information on the items and data collection is available in the Supplement. The six questions were crossed with all three possible pairings of social partners, yielding 18 total items in total that were administered to participants. We again used Bayesian logistic regression to analyze these data, with a standard normal prior designated for regression coefficients and standardized social value scores.

## Reporting summary
Further information on research design is available in the Nature Portfolio Reporting Summary linked to this article.

## Results
### Exploratory phase
#### Assessing links between social value scores and different attitudes about relationship partners
Relationship quality. Across the five exploratory subsamples, social value scores tended to correlate well with relationship quality scores for friends (mean $r = 0.31$, moderate to robust evidence observed in 4/5 subsamples) and acquaintances (mean $r = 0.19$, moderate to robust evidence observed in 3/5 subsamples). The estimates for parent relationship quality were less stable from sample to sample (mean $r = -0.04$, moderate to robust evidence observed in 5/5 subsamples), with some positive correlations observed in some subsamples and negative correlations observed in others. Results broken down by subsample are listed in Table 4.

Social loss aversion towards social partners. Social value scores were generally positively correlated with social loss aversion for all three social partners (mean $r = 0.11$ for parents, friends, and acquaintances; moderate to robust evidence observed in 3 out of 5 subsamples for all three social partners). Results broken down by subsample are listed in Table 4.

Time spent with social partners. The pattern of results involving self-reported actual and ideal time spent with others was less consistent. Social value scores were positively correlated with actual time spent with parents (mean $r = 0.11$, robust evidence observed in 2 out of 5 subsamples) but was weakly negatively correlated with ideal time spent with parents (mean $r = -0.04$). Similarly, the scores were negatively correlated with actual time spent with friends (mean $r = -0.17$, robust evidence observed in 2 out of 5 subsamples) and weakly negatively correlated with ideal time spent (mean $r = -0.02$). The inverse pattern was observed with acquaintances (actual time spent: mean $r = 0.02$; ideal time spent: mean $r = 0.14$, moderate to robust evidence observed in 5 out of 5 subsamples). Results broken down by subsample are listed in table 4.

#### Assessing links between social value scores and social decision preferences among pairs of relationship Partners.
Results of the multiple regression analyses across the exploratory subsamples were noisier than the prior two types of analyses. While the posterior means of regression coefficients generally signified a relationship between social value scores and relevant outcomes in the expected direction (e.g., greater social value scores for parents were associated with greater likelihoods of choosing a parent over a friend or acquaintance in the dictator game and when making forced-choice decisions about with whom to spend time), the posterior variances were quite large (Supplementary Table 3). This could be due to the combination of increased analytic complexity (i.e., including more predictors in the model) and relatively small sample sizes. Because the general trend of the coefficients' directionality was consistent with our hypotheses (i.e., a greater social value score for a social partner was linked to a greater propensity to favor them during decision-making), we were eager to test whether evidence would be more robust in a confirmatory sample with many more participants.

Interim summary. Across five independent subsamples, we found a relatively consistent pattern of results showing that social value scores tracked with other aspects of interpersonal relationships. Social value scores were associated with relationship quality and how upset participants would be if they could no longer spend time with the individual (social loss aversion); greater social value scores for particular social partners tended to be associated with choices favoring those partners in decision-making paradigms. Curiously, associations between social value scores and the total amount of time that people spent, or wished to spend, with a given social partner, were less consistent. Thus, while the method for computing social value used here incorporates information about time spent engaging in particular activities with a social partner, it does not appear to be reducible merely to overall time spent with that person. Coupled with the finding that social value scores were associated with both a validated measure of relationship quality and

**Table 4 | Bivariate correlations between social value scores and several features of participants' relationships with social partners (exploratory sample)**

| Social Value of Target (Subsample) | Relationship Quality | Social Loss Aversion | Actual Time Spent | Ideal Time Spent |
|---|---|---|---|---|
| Parent (Sona-1) | -0.22 [-0.51, 0.05] | -0.01 [-0.10, 0.15] | 0.06 [-0.23, 0.37] | -0.31 [-0.59, -0.05] |
| Friend (Sona-1) | 0.11 [-0.20, 0.39] | 0.09 [-0.19, 0.40] | -0.03 [-0.32, 0.29] | 0.08 [-0.23, 0.38] |
| Acquaintance (Sona-1) | 0.04 [-0.29, 0.36] | 0.22 [-0.06, 0.52] | 0.21 [-0.07, 0.50] | 0.23 [-0.06, 0.52] |
| Parent (Sona-2) | -0.32 [-0.60, -0.02] | -0.09 [-0.41, 0.26] | 0.32 [0.02, 0.63] | 0.01 [-0.32, 0.35] |
| Friend (Sona -2) | 0.41 [0.15, 0.69] | -0.08 [-0.43, 0.23] | -0.50 [-0.75, -0.24] | -0.26 [-0.58, 0.06] |
| Acquaintance (Sona-2) | 0.29 [0.00, 0.60] | 0.14 [-0.16, 0.48] | 0.13 [-0.19, 0.46] | 0.30 [-0.02, 0.59] |
| Parent (Sona -3) | -0.11 [-0.31, 0.08] | 0.03 [-0.17, 0.24] | -0.07 [-0.27, 0.12] | -0.09 [-0.27, 0.11] |
| Friend (Sona -3) | 0.09 [-0.10, 0.28] | 0.00 [-0.20, 0.19] | 0.02 [-0.16, 0.21] | 0.07 [-0.13, 0.25] |
| Acquaintance (Sona -3) | 0.30 [0.14, 0.48] | 0.24 [0.07, 0.44] | 0.10 [-0.10, 0.27] | 0.28 [0.09, 0.45] |
| Parent (MTurk-1) | 0.15 [-0.04, 0.33] | 0.26 [0.09, 0.43] | 0.02 [-0.16, 0.22] | -0.03 [-0.22, 0.15] |
| Friend (MTurk-1) | 0.30 [0.14, 0.48] | 0.20 [0.01, 0.38] | -0.01 [-0.19, 0.18] | 0.13 [-0.03, 0.32] |
| Acquaintance (MTurk-1) | 0.01 [-0.18, 0.20] | -0.05 [-0.24, 0.13] | -0.05 [-0.25, 0.12] | 0.08 [-0.10, 0.28] |
| Parent (MTurk-2) | 0.28 [0.10, 0.48] | 0.35 [0.17, 0.52] | 0.24 [0.03, 0.42] | 0.21 [0.01, 0.41] |
| Friend (MTurk-2) | 0.65 [0.53, 0.77] | 0.32 [0.14, 0.52] | -0.31 [-0.52, -0.14] | -0.14 [-0.35, 0.07] |
| Acquaintance (MTurk-2) | 0.31 [0.13, 0.51] | 0.00 [-0.20, 0.20] | -0.31 [-0.51, -0.12] | -0.17 [-0.37, 0.03] |

*Note.* Brackets represent 89% HDIs of posterior probability distributions. Subsamples are listed in parentheses; 'Sona' or 'MTurk' reflect where the sample completing the likelihood ratings was recruited from. Sona-2 was administered MTurk-sourced activities; MTurk-2 was administered Sona-sourced activities. All other samples were administered activities sourced from another sample within the same population. 'Relationship Quality' refers to mean scores from the IPPA scale. 'Social Loss Aversion' refers to scores on the one-item measure of how upset a participant would feel if they could no longer spend time with a given social partner. 'Actual Time Spent' refers to how many days in a month, on average, a participant sees a given social partner. 'Ideal Time Spent' refers to how many days in a month, on average, a participant would ideally see a given social partner.

**Table 5 | Correlations between social value scores and relationship quality and social loss aversion, respectively (confirmatory sample)**

| Social value of partner (Subsample) | Relationship quality | Social loss aversion |
|---|---|---|
| Parent (Sona_MTurk) | 0.26 [0.09, 0.42] | 0.23 [0.04, 0.39] |
| Friend (Sona_MTurk) | 0.44 [0.30, 0.60] | 0.24 [0.05, 0.41] |
| Acquaintance (Sona_MTurk) | 0.23 [0.05, 0.40] | 0.09 [-0.08, 0.27] |
| Parent (MTurk_MTurk) | 0.17 [0.02, 0.34] | 0.22 [0.07, 0.38] |
| Friend (MTurk_MTurk) | 0.22 [0.06, 0.39] | 0.26 [0.09, 0.41] |
| Acquaintance (MTurk_MTurk) | 0.16 [-0.00, 0.32] | 0.28 [0.13, 0.44] |
| Parent (Sona_Prolific) | 0.19 [0.09, 0.29] | 0.15 [0.03, 0.26] |
| Friend (Sona_Prolific) | 0.28 [0.19, 0.39] | 0.17 [0.06, 0.28] |
| Acquaintance (Sona_Prolific) | 0.37 [0.27, 0.47] | 0.34 [0.24, 0.43] |
| Parent (MTurk_Prolific) | 0.06 [-0.05, 0.17] | 0.15 [0.03, 0.26] |
| Friend (MTurk_Prolific) | 0.09 [-0.02, 0.20] | 0.15 [0.04, 0.26] |
| Acquaintance (MTurk_Prolific) | 0.18 [0.06, 0.28] | 0.24 [0.13, 0.34] |

*Note.* Brackets represent 89% HDIs of posterior probability distributions. Subsamples are listed in parentheses; the first term denotes the population from which the activities were sourced; the second term denotes which population completed the likelihood ratings and other measures. 'Relationship Quality' refers to mean scores from the IPPA scale. 'Social Loss Aversion' refers to scores on the one-item measure of how upset a participant would feel if they could no longer spend time with a given social partner.

participants' choices regarding who they would be most upset not to be able to spend time with lends confidence to the validity of the method of computing social value. We sought to replicate and extend these findings in a larger sample for the confirmatory phase of the study.

## Confirmatory phase
Results from this phase of the study are divided into two sections: replication and expansion. The replication section contains analyses that attempt to directly replicate and confirm the results of the exploratory phase. The expansion section contains analyses that aim to further characterize the relationship between social value and social behavior by assessing relationships between social value scores and two additional sets of outcomes (multi-trial social decision preferences and one-off choices about affiliative social behavior).

**Replication.** Below we report replications of the results regarding relationship quality and social loss aversion from the exploratory phase of the study. Questions regarding time spent with each partner were dropped to reduce task demands after expansion measures were added.

### Assessing links between social value scores and attitudes about relationship partners
Relationship quality. Replicating findings from the exploratory sample, social value scores were positively correlated with relationship quality for both friends and acquaintances (friend: mean $r = 0.26$, moderate to robust evidence observed in 4/4 subsamples; acquaintance: mean $r = 0.24$, moderate to robust evidence observed in 4/4 subsamples). In this phase of the study, social value scores were also positively correlated with parent relationship quality over all the subsamples (mean $r = 0.17$, moderate to robust evidence observed in 4/4 subsamples). Results for each subsample are listed in Table 5. See Fig. 2 for visualization of these results.

Social loss aversion towards social partners. Replicating results from the exploratory phase of the study, social value scores were correlated with social loss aversion (parent: mean $r = 0.19$, robust evidence observed in 4/4 subsamples; friend: mean $r = 0.21$, robust evidence observed in 4/4 subsamples; acquaintance: mean $r = 0.24$, moderate to robust evidence observed in 3/4 subsamples). Results for each subsample are listed in Table 5. See Fig. 2 for visualization of these results.

### Assessing links between social value scores and social decision preferences among pairs of relationship partners.
Multiple regression analyses in this phase yielded clearer results in the anticipated direction. Individual differences in social value scores tracked with choice preferences on the dictator game in the expected direction for all three social partner pairings, indicating that social value scores can be used to predict

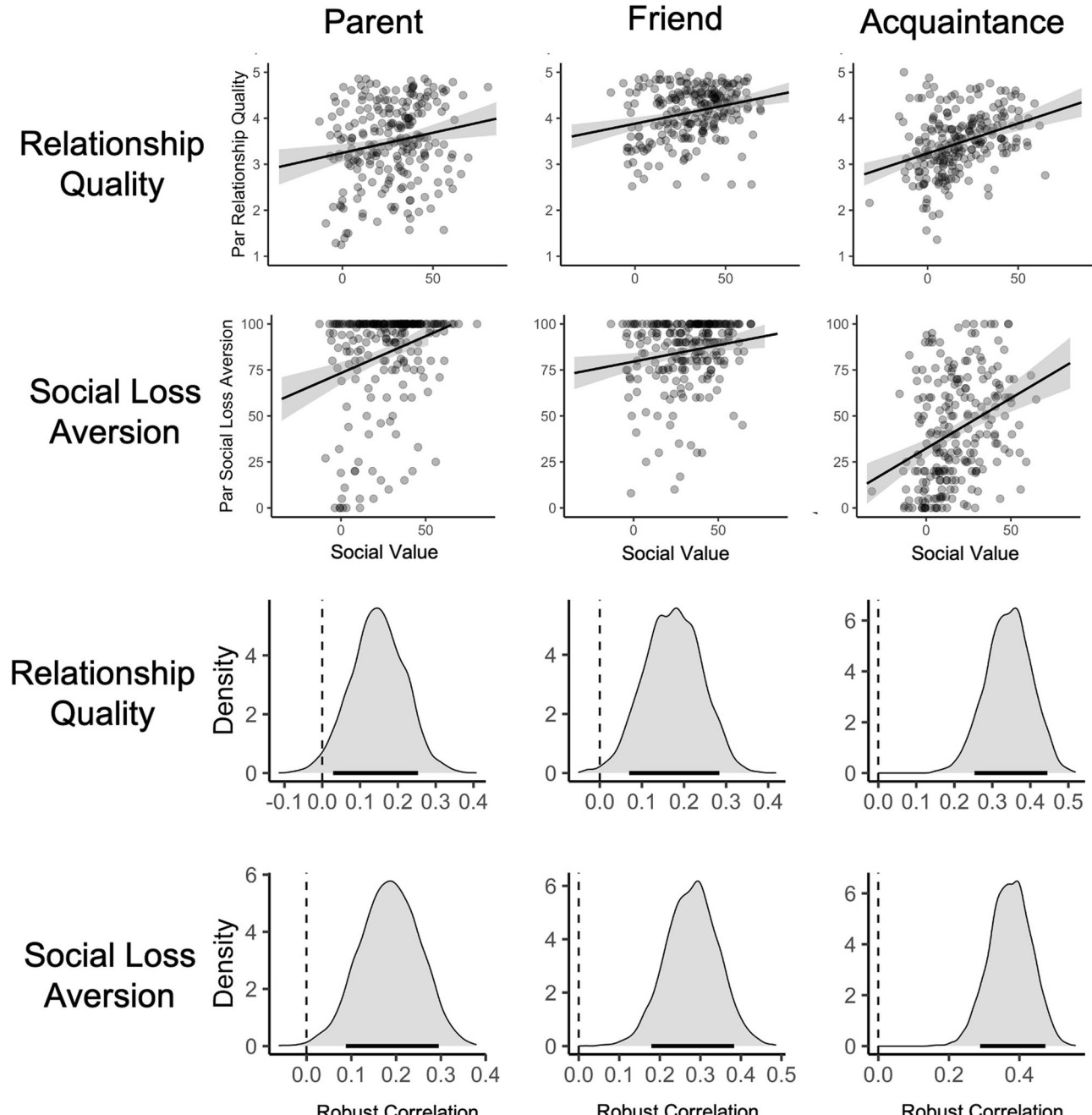

**Fig. 2 | Social value scores are correlated with relationship quality and social loss aversion.** Data for this visualization were drawn from confirmatory phase (prolific sample, SONA-sourced weights). The top set of scatter plots visualizes relationship between social value scores and outcome variables for a given social partner. Pearson correlations are visualized with the trend lines in the top set of plots. The bottom set of plots depicts the posterior distributions for each correlation (the likelihood of observing a given correlation value given the data). Posterior distributions were obtained via MCMC sampling. The black bars underlying each posterior distributions represents the 89% highest density credible interval (HDI).

A hashed vertical line is shown over zero (null value). Evidence was judged to be robust if the HDI did not include 0 or the HDI fell outside of the Region of Practical Equivalence (ROPE) and moderate if part of the HDI fell outside of ROPE (see "Inferential Criteria" section of the main text). ROPE was defined as the range between −0.1 and 0.1. Here, there was robust evidence for an association between social value scores and relationship quality for all three social partners. There was also robust evidence for an association between social value scores and social loss aversion for all three social partners. The sample size for this analysis is $N = 233$.

whom an individual will favor when making conflicting choices involving social partners.

Results from the forced choice question about spending time with one of two social partners yielded similar results, albeit with somewhat less consistency. Nevertheless, social value scores in this context were also able to predict whom an individual would prefer to spend free time with. One exception to this, however, pertained to the friend–acquaintance pairing; regression coefficients from this pairing were observed in the predicted

direction but posterior variances were quite high. Nevertheless, we take the direction of the coefficients to be encouraging in conjunction with other results. Full results can be accessed in Supplementary Table 4. See Fig. 3 for visualization of these results.

**Interim summary**. Results in this phase of the study show a more consistent relationship between social value and relationship quality, social loss aversion, and social decision-making. This is likely due to the

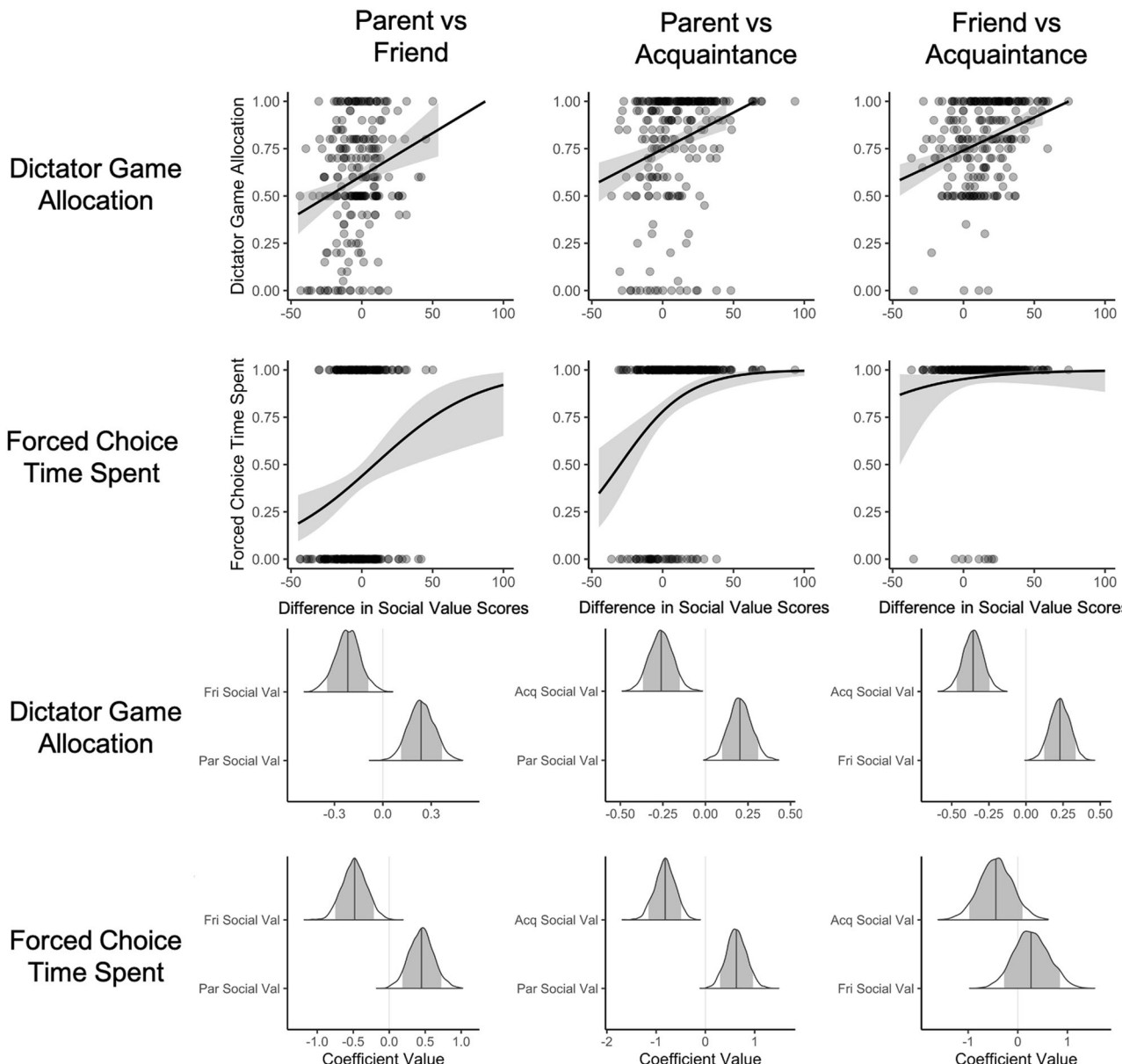

**Fig. 3 | Social value scores predict choice preferences for specific social partners in one-shot social decision tasks with monetary and social outcomes.** Analyses involving this set of outcome variables were conducted by regressing each outcome onto the relevant pair of social value scores. For simplicity in visualization, figures in the top two rows plot the relationship between difference scores between social value score pairs and each outcome. 'Forced Choice Spend Time' refers to a one-shot question asking participants to choose one of two social partners with whom they would rather spend a free afternoon. Data for this visualization were drawn from confirmatory phase of the study (prolific sample, SONA-sourced weights). A greater value on the y-axis of the plots in the top two rows indicates greater preference for the first social partner listed in the axis label. Axis labels reflect differences between social value scores. The bottom two rows of plots depict the posterior distributions (the likelihood of observing a given coefficient value given the data) for each slope coefficient from the full multiple regression model. Posterior distributions were obtained via MCMC sampling. The shaded portion of the posterior distributions represent the 89% highest density credible interval (HDI). A hashed vertical line is shown over zero (null value); a solid vertical line within each posterior indicates the posterior mean. Evidence was judged to be robust if the HDI did not include 0 or the HDI fell outside of the Region of Practical Equivalence (ROPE) and moderate if part of the HDI fell outside of ROPE (see "Inferential Criteria" section of the main text). ROPE was defined as the range between -0.1 and 0.1. Here, there was robust evidence that social value scores predict choice preferences involving monetary outcomes for all social partners. There was also robust evidence that social value scores predict choice preferences involving social outcomes when pitting friends or acquaintances against parents. There was moderate evidence that social value scores predict choice preferences involving social outcomes when pitting friends against acquaintances. A logistic-link function was used for analyses with the binary forced choice item about spending time with a social partner. 'Acq' refers to acquaintance, 'Fri' refers to friend, 'Par' refers to parent, and 'Val' refers to the computed social value score. The sample size for this analysis is $N = 233$.

larger sample sizes for each confirmatory subsample, allowing for more stable estimation of individual differences. We next examined the relationship between social value scores and additional measures of social decision-making to further characterize the relationship between estimates of targets' social value and social behavior toward those targets.

### Expansion

**Multi-trial social decision-making paradigm.** Participants completed several runs of an existing multi-trial social decision-making paradigm[47]. Similar to our one-shot questions, the paradigm presented participants with conflicting options for allocating money or time between all three possible pairings of social partners (one pair per trial). Results are shown in Fig. 4 and

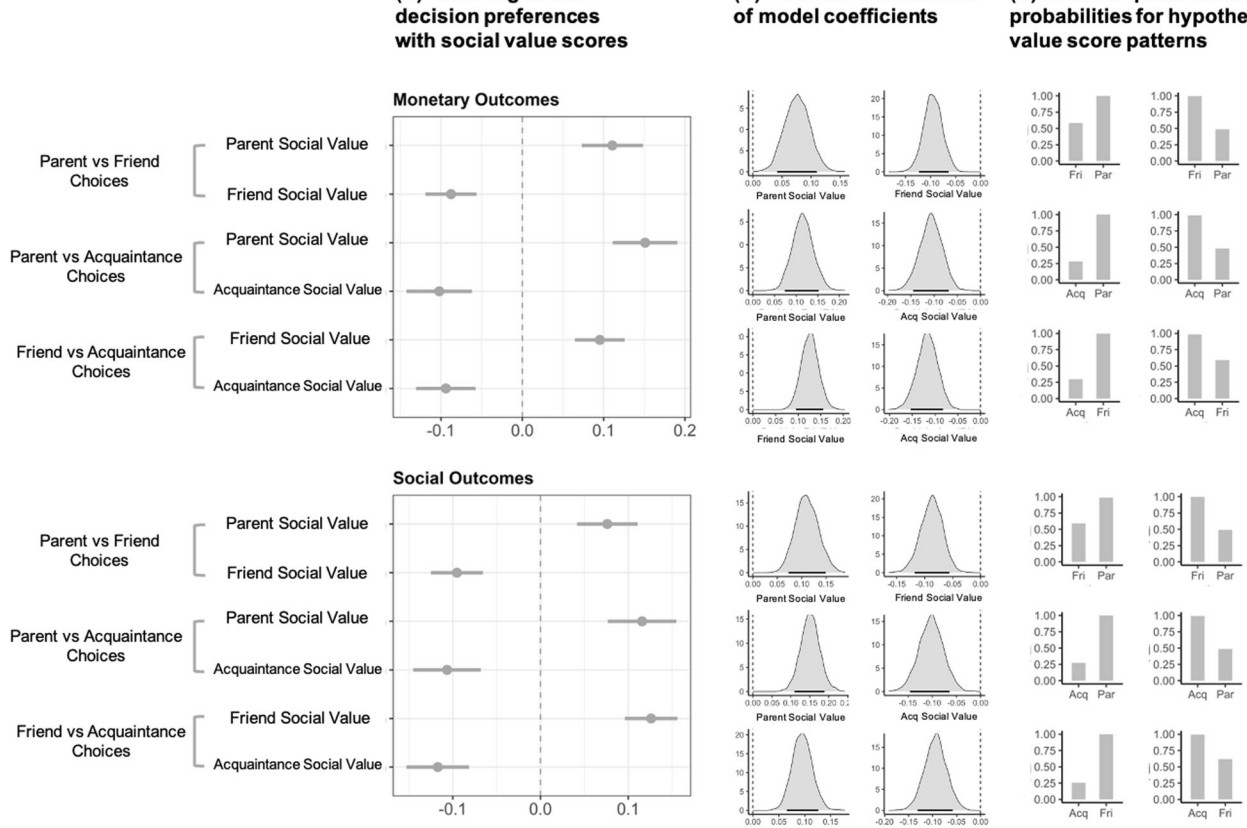

**Fig. 4 | Social value scores predict choices in a multi-trial social decision paradigm. A** The left column contains dot-and-whisker plots showing the association between social value scores and social decision preferences for a given social partner in a given condition. The negative coefficients in these plots are expected. **B** The center column depicts posterior distributions of these effects. Both the left and center columns depict 89% high density credible intervals (HDIs). Evidence was judged to be robust if the HDI did not include 0 or the HDI fell outside of the Region of Practical Equivalence (ROPE) or moderate if part of the HDI fell outside of ROPE (see "Inferential Criteria" section of the main text). ROPE was defined as the range between −0.1 and 0.1. Here, there was robust evidence that social value scores predict choices for all social partners for both monetary and social outcomes in a multi-trial social

decision paradigm. **C** The right column displays bar plots depicting the model-implied probabilities of favoring a given social partner for social value scores 1.5 SDs above the mean for a given social partner and 1.5 SDs below the mean for another given social partner (left sub column, other 1 is 1.5 below, other 2 is 1.5 above; opposite for right sub column). Data for this visualization were drawn from confirmatory phase (prolific sample, SONA-sourced weights). 'Par' refers to parent, 'Fri' refers to friend, and 'Acq' refers to acquaintance. 'Soc Val' refers to social value scores. 'Density' refers to the mass of the posterior distribution. Posterior distributions were obtained via MCMC sampling. The black bars underlying each posterior distributions represents the 89% highest density credible interval. A hashed vertical line is shown over zero (null value). The sample size for this analysis is $N = 233$.

in Supplementary Tables 5–7. All effects subsequently reported here remain the same when statistically adjusting for relationship quality. Results are reported below, broken down by relationship type pairings. Comprehensive statistical modeling details are included in the Supplement.

Parents vs friends. When considering choices between close others (i.e., parents and friends), greater social value scores for parents were generally associated with stronger parent-over-friend preferences, regardless of outcome type (i.e., social or monetary) and activity pool (i.e., sourced from SONA or MTurk participants). The same was true for friend social value scores, such that greater social value scores for friends were associated with stronger friend-over-parent preferences across outcome types and activity pools. These findings in independent samples from Prolific and MTurk. See Fig. 4 for visualizations of results involving this outcome.

Close others vs acquaintances. We next examined choices between close others (parents, friends) and acquaintances. When considering choices between parents and acquaintances, social value scores for parents again tracked with the likelihood of favoring the parent on the task (again, regardless of outcome type or activity pool). This same pattern of findings observed in the larger Prolific sample was also observed in the smaller Mturk

sample (see Materials and Methods). Similarly, in both the Prolific and Mturk samples, when considering choices between friends and acquaintances, social value scores for friends tracked with the likelihood of making decisions favoring friends across outcome types and activity pools. For both parent vs acquaintance and friend vs acquaintance decisions, findings regarding acquaintances' social value scores were less consistent than findings regarding close others' social value scores, particularly in the smaller Mturk subsamples. See Fig. 4 for visualizations of results involving this outcome.

Preferences for social affiliative behaviors. Social value scores were consistently predictive of participants' preferences regarding affiliative behaviors for questions that pitted parents against friends; regression coefficients were in the expected directions for both parent (positive) and friend (negative) social value scores and posterior distributions evinced moderate to robust evidence for the majority of the items across all four confirmatory subsamples. The same was generally true for items pitting parents against acquaintances, as well as those pitting friends against acquaintances, although the evidence was somewhat less consistent. Nevertheless, taken together, these results highlight that social value scores are able to predict preferences regarding common social affiliative behaviors. Full results can be accessed in Supplementary Tables 7-10.

Finally, we again examined paired differences in social value and relationship quality among the relationship types. These results for the confirmatory sample are provided in the Supplement (Supplementary Table 11).

## Post hoc analyses: further psychometric assessment of social value scores

After completing the exploratory and confirmatory analyses, we performed additional post-hoc analyses to help gauge the validity of social value scores. This included verifying the reliability of the likelihood ratings by (i) calculating the internal consistency of likelihood ratings and (ii) testing whether weights from the behavioral signature were simply a source of noise, (iii) checking to see whether social value scores were still statistically related to key variables (relationship quality, social decision behavior, etc.) when controlling for unit-weighted likelihood scores (i.e., do social value scores predict outcomes over and above the likelihood scores alone), (iv) verifying whether social value scores had discriminant validity (i.e., were not associated with things that they logically should not be associated with), and finally, (v) testing whether social value scores were predictive of social behavior (e.g., social decision preferences, social affiliative behaviors) over and above other ways of assessing social relationships (e.g., relationship quality, social loss aversion). We conducted these analyses on two of our confirmatory phase samples: one with SONA-sourced activities and the other with MTurk sourced activities (both collected from the Prolific platform). These samples were chosen because they had the largest sample sizes. These analyses and their results are described in detail below.

**Internal consistency**. In order for social value scores to be useful, they must be reliable. One way to begin to evaluate the reliability of said scores is to determine the internal consistency of the likelihood ratings, a key substrate of social values scores. Before doing so, there are two related points to be kept in mind. First, unit-weighted (i.e., sum or mean) likelihood scores are not measuring the same construct as our social value scores. Second, and relatedly, measures of internal consistency typically evaluate the reliability of composite scores assuming unit weighting of instrument items. Because our social value scores are not derived from unit-weighted likelihood ratings, it is important to keep in mind that the reliability of the likelihood ratings are only *part* of the evaluation for the reliability of social value scores, and that relying solely on the application of classic psychometric techniques to likelihood ratings can lead to potentially misleading conclusions. However, ascertaining the reliability of likelihood ratings is still necessary for establishing the reliability of social value scores.

With this in mind, we first computed model-based reliability of likelihood ratings from the exploratory and confirmatory samples for both the parent, friend, and acquaintance ratings. We focused on model-based indices of reliability—$\omega_{total}$, $\omega_{hierarchical}$, and explained common variance (ECV). These indices are computed from bi-factor models that decompose covariance between observed indicators as a function of a general latent factor—contributing to all indicators—and several group factors that contribute to a subset of items. $\omega_{total}$ captures the variance accounted for by all latent factors, $\omega_{hierarchical}$ captures the amount of total variance captured by the general factor, and the ECV captures how much of the common variance is accounted for by the general factor. The first index is a measure of the overall reliability of an instrument, whereas the latter two capture the general factor saturation. That is, some instruments may be multi-dimensional in theory, but may be unidimensional in practice due to having a strong common factor. We computed these indices using the *omega* function from the psych R package. For sensitivity, we varied the number of group factors[3–5]. Likelihood ratings displayed internal consistency and strong general factor saturation both in the sample with SONA-sourced activities (*Parent*: $\omega_{total}$ 0.982–0.983, $\omega_{hierarchical}$ 0.759–0.883, ECV 0.592–0.741; *Friend*: $\omega_{total}$ 0.982–0.984, $\omega_{hierarchical}$ 0.732–0.786, ECV 0.567–0.635; *Acquaintance*: $\omega_{total}$ 0.987–0.989, $\omega_{hierarchical}$ 0.774–0.808, ECV 0.641–0.679) and in the sample with MTurk-sourced activities (*Parent*:

$\omega_{total}$ 0.972–0.975, $\omega_{hierarchical}$ 0.651–0.700, ECV 0.493–0.552; *Friend*: $\omega_{total}$ 0.972–0.975, $\omega_{hierarchical}$ 0.681–0.708, ECV 0.484–0.552; *Acquaintance*: $\omega_{total}$ 0.984–0.985, $\omega_{hierarchical}$ 0.775–0.799, ECV 0.637–0.703). These results suggest excellent reliability and a practically unidimensional measure for all three types of relationships that were assessed here.

As part of this process, we also tested whether the activity weights evinced measurement invariance between social partners. We focused on testing for configural invariance, which establishes equivalence in the factor structure, because other forms of measurement invariance involve factor loadings and intercepts and these quantities are not relevant here because we are not unit-weighting each individual item. We found moderate evidence for configural invariance, as indexed by RMSEA (root mean squared error of approximate) absolute fit statistics (all values between .07 and .08 in all confirmatory subsamples). However, using the comparative fit index (CFI) revealed statistics that were very poor (in 0.42–0.53 range). One explanation for this could be due to the fact that while the factor structure is *essentially unidimensional* for all three sets of social partners independently (based on the high $\omega_{hierarchical}$ values), a strict test of this assumption fails. We view this as important to test in future work but remain cautiously optimistic in the interim. We could not test for invariance based on age (college age vs non-college age) or online sample (SONA vs MTurk) due to sample size constraints.

**Confirming the utility of behavioral signature weights**. To further verify the reliability of the social value scores, we checked to see whether the weights from the behavioral signature were meaningfully contributing to the calculation of social value scores. Said differently, we checked to make sure that the link between an activity and its weight was not arbitrary. To do this, we randomly shuffled weights of the behavioral signature, computed social value scores with the shuffled weights, and then re-ran several statistical analyses involving social value scores. We specifically re-ran correlations with relationship quality and social loss aversion, as well as paired differences between social value scores of each known other. Other analyses were not re-run in this test given computational demands.

Across both confirmatory datasets used here, we consistently observed that repeatedly shuffling the pairing between weights and likelihood ratings for a given activity and then re-computing standardized test statistics of interest resulted in distributions largely centered around a null value. This was true in the confirmatory sample with SONA-sourced activities for both for correlations between social value scores and relationship quality (*Parent*: $r_{shuffled} = -0.039$ (SD = 0.14), *Friend*: $r_{shuffled} = -0.022$ (SD = 13), *Acquaintance*: $r_{shuffled} = -0.040$ (SD = 0.14)) and for correlations between social value scores and social loss aversion (*Parent*: $r_{shuffled} = -0.036$ (SD = 0.14), *Friend*: $r_{shuffled} = -0.020$ (SD = 0.10), *Acquaintance*: $r_{shuffled} = -0.054$ (SD = 0.13)). This was also true in the confirmatory sample with SONA-sourced activities, both for correlations between social value scores and relationship quality (*Parent*: $r_{shuffled} = -0.098$ (0.11), *Friend*: $r_{shuffled} = -0.069$ (SD = 0.11), *Acquaintance*: $r_{shuffled} = -0.141$ (SD = 0.14)) and for correlations between social value scores and social loss aversion (*Parent*: $r_{shuffled} = -0.085$ (SD = 0.14), *Friend*: $r_{shuffled} = -0.069$ (SD = 0.13), *Acquaintance*: $r_{shuffled} = -0.152$ (SD = 0.13)). A similar pattern of null results was found when examining paired differences between social value scores for each known other using shuffled weights, both in the confirmatory sample with SONA-sourced activities (*Parent–Friend*: $d_{shuffled} = 0.033$ (SD = 0.37), *Parent–Acquaintance*: $d_{shuffled} = -0.018$ (SD = 0.39), *Friend–Acquaintance*: $d_{shuffled} = -0.056$ (SD = 0.26)) and in the confirmatory sample with MTurk-sourced activities (*Parent–Friend*: $d_{shuffled} = 0.087$ (SD = 0.33), *Parent–Acquaintance*: $d_{shuffled} = -0.090$ (SD = 0.41), *Friend–Acquaintance*: $d_{shuffled} = -0.185$ (SD = 0.27)). The one mild exception to the tendency for this approach to yield test statistics centered on a null value involved statistics related to acquaintance scores in the sample with MTurk-sourced weights, which were slightly elevated (around ~0.1 for $d$ and $r$ values) in the *opposite* direction. While we point this out in the interest of full disclosure, we are unconcerned with this

finding because these effect sizes are still quite small and the variance of the shuffled distributions around them is still quite large.

**Social value scores are still largely related to behavior when controlling for unit-weighted likelihood ratings.** Another potential issue with our method for evaluating social value is that the associations between social value scores and constructs of interest could be solely driven by the likelihood ratings. To verify that this was not the case, we conducted several Bayesian multiple regression analyses where we entered social value scores and unit-weighted mean likelihood ratings into the same model predicting each outcome of interest. All predictors were standardized and a weakly informative prior was used for slope coefficients (standard normal, $N(0,1)$). Here we examined four outcome variables: relationship quality, social loss aversion, dictator game choice preferences, and choice preferences on the forced choice question about spending time with one of two others.

Results are listed in Supplementary Tables 12–13. Results for relationship quality and social loss aversion across both samples evinced the same consistent pattern of findings: the vast majority of the posterior distributions for the slope coefficient corresponding to social value scores fell in the anticipated direction. Concretely, this means the 89% HDI did not overlap with zero, or narrowly did so. The magnitude of slope coefficients for the unit weighted likelihood ratings was consistently greater than the social value coefficients.

In both confirmatory samples, social value scores were consistently associated with choice preferences on the dictator game. That said, social value scores were largely unrelated to the forced choice question about spending time with one of two social partners in both confirmatory samples when controlling for unit-weighted likelihood scores. This suggests that one's overall willingness to spend time with someone, as indicated by the likelihood ratings component of social value scores, accounts for the statistical association between social value scores and forced choice decisions about whom to spend time with. While this result logically makes sense given the nature of the particular outcome variable at hand, it differs from what was observed in analogous analyses with other variables, where social value scores remained associated with relationship quality, social loss aversion, and choice preferences in the dictator game after controlling for unit-weighted likelihood scores.

**Social value scores show good discriminant validity.** We next tested whether social value scores showed good discriminant validity (i.e., they were *not* correlated with theoretically unrelated constructs). To do this, we correlated social value scores for a given social partner with relationship quality and social loss aversion for all other partners. This test operates under the assumption that social value scores estimated for a given relationship (e.g., one's acquaintance) should not be substantially predictive of relationship features (e.g., relationship quality, social loss aversion) of one's *other* relationships. This meant that we ran 12 tests in each of the three samples (3 partners' social value scores x 2 mismatched social partners x 2 outcome variables). To contextualize the correlation coefficients and judge discriminant validity, we adopted an empirical effect size heuristic threshold of $r = 0.12$ recently derived from the social psychology literature[48].

Overall, social value scores evinced good discriminant validity. Ten of the twelve correlations in the confirmatory sample with SONA-sourced activities resulted in correlation coefficients at or below 0.12. The two correlations exceeding the were parent social value score associations with friend relationship quality ($r = 0.309$) and acquaintance relationship quality ($r = 0.198$). All twelve correlations in the confirmatory sample with MTurk-sourced activities were estimated to be less than or equal to 0.12 and thus none reached the threshold for a medium effect size.

Following up on the correlations that exceeded the threshold, it occurred to us that those correlations may be driven by individual differences in how individuals place social value on relationships more generally. We reasoned one way to address this would be to re-compute the correlation

between each pair of variables after partialing out the variance of social value scores belonging to the second person (e.g., parent social value scores were correlated with friend relationship quality after partialing out variance from friend social value scores). When doing so, the value of both correlations dropped substantially ($r_{partial}$ [parent social value, friend relationship quality]: 0.198, $r_{partial}$ [parent social value, acquaintance relationship quality]: 0.077). This suggests some of the covariance in these associations is attributable to non-relevant individual differences.

**Are social value scores predictive of behavior when controlling for other facets of relationships?**. Having further established the reliability and validity of social value scores, we next checked whether social value scores are predictive of behaviors when controlling for other facets of relationships. Given our available data, this meant regressing social choice preferences on the dictator game or the forced choice question about spending time with one of two social partners onto social value scores, relationship quality, and social loss aversion. Because each of the two dependent values requires participants to decide between a pair of known others (e.g., parent versus friend, friend versus acquaintance), we differenced the relationship facets and social value scores before entering them into the model to for better interpretability. To illustrate this, we ran one model in which we subtract relationship facets and social value scores for friend from parent and then entered the differences into a model predicting allocations between parent and friend on the dictator game. We enacted this procedure in the three aforementioned samples using Bayesian regression (all predictors were standardized and received standard normal priors).

Results from the models are depicted in Supplementary Figs 3–4. Overall, social value scores showed modest but consistent associations with outcomes. While we generally found moderate evidence for these associations (i.e., part of each HDI fell outside of ROPE; see "Inferential Criteria"), and most of the posterior mass in many of these analyses fell around either side of zero (i.e., overlapped with zero). Some associations between social value and choice behavior controlling for other relationship facets were stronger, but these results were inconsistent. These analyses suggest that social value is a modest yet persistent predictor of social behavior after adjusting for other facets of social relationships.

## Discussion
The goal of this study was to develop and validate a behavioral signature of social value that could be applied to quantify the value of interpersonal relationships. Using a definition of social value rooted in economic concepts of scarcity and opportunity cost, we devised and implemented a method that calculates the social value of interpersonal relationships by examining how likely individuals are to complete highly prioritized activities with a given relationship partner. Our method of engineering a behavioral signature of social value produced scores that tracked with self-reported relationship quality, social loss aversion, and patterns of behavior towards specific social partners on social decision-making tasks. Crucially, these findings do not appear to be driven by mere time spent (actual or ideal) with familiar others, as our social value scores were largely uncorrelated with these metrics. Our results have wide-ranging implications for theories in the behavioral sciences.

These results complement the vast literature on value-based human behavior. Prominent theories posit that human behavior is guided by cognitive and affective heuristics that compute the subjective value of prospective actions, and the action that maximizes subjective value is then enacted[49–53]. These theories thus place an emphasis on the subjective value of individual *actions*. Our work could help enrich these theories by showing that individuals assign value to specific *agents* in their social environment. This would allow for a more comprehensive understanding of how different elements of one's environment shape value appraisals for specific actions. This promises to provide additional mechanistic specificity to these theories by enabling researchers to partition the value of a prospective action into its constituent components (e.g., how much of a given action's value comes

from the value of the relationship partner or from other features of the action). The potential for enhanced mechanistic specificity is not only relevant for basic theory, but also has practical applications, such as for advancing understanding of how value-based processes are disrupted in psychiatric illnesses.

Our findings are also relevant for basic theories of human relationships. Much scientific work had been devoted to measuring interpersonal relationship quality[54–57]. While such measures are important and predictive of many significant outcomes[15,58,59], we reasoned that characterizing the value that people ascribe to their social relationships would benefit from incorporating information about the ways that people choose to spend their limited time when cultivating and maintaining those relationships. Indeed, our results suggest that interpersonal relationship value is related to, but ultimately distinct from, relationship quality, as well as several other ways of evaluating relationships. While our findings also show that social value is a motivator of social behavior, our *post hoc* analyses lead us to conclude that other ways of assessing relationships (social loss aversion, relationship quality) are stronger predictors of the behavioral measures used here. Given that our method for calculating social value appears to yield reliable and valid scores, we interpret this finding as helping delineate the facets of social relationships that likely shape social behavior. Our evidence suggests that social value has a small effect in shaping behavior, but is partially eclipsed by relationship quality and social loss aversion. These findings are non-trivial in our view because they help untangle the nomological network of associations between relationship facets and social behavior.

Relatedly, the pattern of findings observed with social loss aversion—a measure that we created here in this study intended to help validate social value scores—is also of interest. Our social loss aversion metric is similar to relationship quality in that it taps an attitude or sentiment towards a specific social partner, but differs in that it frames the sentiment in terms of anticipated *negative affect* over losing access to said social partner. Given well documented asymmetries in value-based processes between gains and losses[53], we find it compelling that social loss aversion was as predictive of social decision preferences, if not more so in some cases, than relationship quality. This finding could represent an intriguing breadcrumb trail for future work, as it could yield insights into how asymmetric value-based processing of gains and losses apply to social contexts while opening up new dimensions to characterize close relationships[60]. An alternate conceptualization of social loss aversion is that it measures another facet of interpersonal relationship value, since the measure is also rooted in economic concepts of scarcity. To us, this is further evidence suggesting it is worthwhile to think about interpersonal relationship value in these economic terms.

The current approach for computing social value focuses primarily on how people prioritize and choose to engage in various behaviors with others. While this approach generated estimates of social value that effectively predicted people's actual behavior in decision-making tasks and their scores on relevant self-report measures, it does not consider all possible features that might contribute to social value. For example, the value that one ascribes to a given social partner could also be shaped by how that person serves one's goals. Indeed, interpersonal relationships have been theorized as means to individual goals[61–63], whether they be economic (e.g., financial security), biological (e.g., having children), or psychological (e.g., companionship). Thus, in the future, the current method could be extended by incorporating information about how people prioritize different goals and the relevance of various behaviors to those goals. Correspondingly, our approach could also be applied to provide enriched mechanistic specificity for such theories of interpersonal relationships that emphasize relationship partners as means to individual goals. Explicitly assigning value to relationship partners may be one way to more formally quantify these theories. We emphasize that our method is intended to be a first step in attempting to define and measure the value of one's specific interpersonal relationships. We acknowledge that it is necessary to develop and evaluate other approaches that 'stress test' some of the assumptions that were made here (e.g., approaches that take into account that the relationship between the social value and activities with others is inevitably bidirectional, subject to contextual factors, and so on).

Finally, another direction for future work involves examining the generalizability of this approach, including generalizability across cultures, as our operationalizations and conceptions of both value and interpersonal relationships, as well as the populations sampled, are inextricable from the culture in which they were studied. Relatedly, we note that here we put forth a *method* for engineering a behavioral signature of social value, not necessarily a generalizable signature, as activity weights from the signatures derived here likely reflect idiosyncrasies of sample characteristics from which they were obtained. Future work could attempt to develop a widely generalizable signature–e.g., by focusing on deriving activity weights that are generalizable across a broader population or by using weights specific to each participant. Particular care will need to be taken when implementing this method in different cultures. While we currently believe that the method, though not necessarily the signatures derived here, are likely to generalize to other WEIRD samples (Western, Educated, Industrialized, Rich, Democratic[64];), this may not be the case for non-WEIRD cultures. Future work seeking to implement this method should be aware of cultural dynamics that may complicate the applicability of the procedure proposed here. In particular, we could foresee a case where cultural norms around social activities in other countries more strongly dictate the acceptability of completing an activity with specific social partners, limiting the degree to which participants could willfully choose how to spend their time in social contexts. In such a case, new items that meaningfully track with the expenditure of finite social capital would need to be identified and used instead of activities completed in one's leisure time.

With this in mind, we note that a strength of our method is the relative ease of implementation, especially after signature weights have been derived. The activity sourcing procedure requires relatively few participants since most activities volunteered by participants are easily collapsed into superordinate categories. The process of paring down activities was quickly accomplished and verified by members of the research team, and could perhaps be automated via large language models in the future. Further, whether it be implementing the MaxDiff design to obtain data to compute activity weights that comprise the signature or administering likelihood ratings of the activities themselves, research processes related to engineering or applying the signature can easily be implemented in relatively little time. Last, administering the likelihood ratings can be achieved quickly: We analyzed survey duration data and estimate that it typically takes ~4.94–6.75 min to complete said ratings for one target individual. Coupled with the fact that the same signature weights can be applied repeatedly to novel samples from a population, we believe that this 'behavioral signature' approach to calculating social value could mirror the utility of comparable approaches in cognitive neuroscience[39].

## Limitations

While we believe the current contribution is notable for the reasons discussed above, additional work is needed to further develop the concept of interpersonal social value. In particular, future studies could continue to map the correlates of social value. It is imperative to determine how well social value tracks with real-world social behavior. Future work could test this by employing ecological momentary assessments to test whether social value is associated with daily social behavior involving specific others. Moreover, if our social value scores are truly indexing value, they should correlate with value-based processes in the human brain. This could be tested by correlating social value scores with the strength of value-based signals elicited in the brain (e.g., via functional magnetic resonance imaging recordings) when viewing or making decisions about specific social partners.

We also envision several important extensions to be pursued in future studies, such as modulating likelihood ratings by anticipated or past enjoyment of completing a given activity with a particular social partner, incorporating information about how being accompanied to unenjoyable

activities by certain social partners can heighten the experience, personalizing the activity weights, fine-tuning them to better reflect relationship by activity interactions, and incorporating the effort cost of each activity in the signature. Of these potential extensions, we see two as particularly important. The first involves the use of personalized or quasi-personalized activity weights. Although the inclusion of high-variance weights in the behavioral signature did not make a difference in terms of the rank ordering of participants' social value scores (see Supplement), future versions of this approach to calculating social value would likely benefit from additional precision if personalized weights were incorporated in some way. Doing so would mitigate any noise that is contributed by inconsistency across participants in their activity preferences. If it is not feasible to complete the weight-generating procedure for each individual participant, then researchers could adopt a quasi-personalized method where a single precomputed weight set is used (as we do here), and low-variance activity weights are retained from the overall model estimate but high-variance activity weights are replaced with participant-specific weights from the precomputed participants by matching individuals from the pre-computed sample and the researcher's focal sample on relevant demographic (e.g., age, sex) or psychological (e.g., personality traits) dimensions by using propensity score matching or another approach. The second significant potential extension involves better incorporating the effort of each activity that comprises the behavioral signature. Addressing this current limitation is important because inadvertent conflation with effort may dilute or bias the calculation of social value. This could be addressed by modifying future prompts when collecting data for calculating activity weights or likelihood ratings, or obtaining independent ratings of effort for each activity and factoring them into the calculation.

## Conclusions

In sum, we developed a method for quantifying the value that one places on specific interpersonal relationships based on how individuals choose to engage with said relationship partners. Value scores derived using this method were correlated with relationship quality as well as several indices of social behavior, but were not merely reducible to the raw amount of time that one spends or wishes to spend with others. It is our hope that these results elucidate another facet of value-based processes in humans–specifically, how people ascribe value to their interpersonal relationships.

## Data availability

All data for this manuscript are freely available on the Open Science Framework at https://osf.io/rjqpc.

## Code availability

All code for this manuscript is freely available on the Open Science Framework at https://osf.io/rjqpc.

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

## Acknowledgements

Preparation of this research was supported by a National Science Foundation SBE Postdoctoral Research Fellowship to JFGM (award number 2104629) and the UCLA Department of Psychology. The funders had no role in study design, data collection and analysis, decision to publish or preparation of the manuscript. We thank L. Concepción Esparza for help in paring down the sourced activities into slightly higher levels of abstraction. We appreciate Drs. Andrew Fuligni, Adriana Galván, Jennifer Silvers, and Tor Wager for a stimulating discussion that inspired the premise of this work. We thank the members of the Computational Social Neuroscience Laboratory at UCLA for their feedback on study concept and development, and for critiquing a draft of this manuscript.

## Author contributions

J.F.G.M. and C.P. developed research questions and designed the study concept. Data were collected, cleaned, and analyzed by J.F.G.M. under the supervision of C.P. J.F.G.M. drafted the manuscript with substantial feedback and critical edits from C.P. Both authors approved the final manuscript for submission.

## Competing interests

The authors declare no competing interests.
