## [Peer Review File · Communications Psychology]

Decision letter and referee reports: first round

Dear Dr Guassi Moreira,

Thank you for your patience during the peer-review process. Your manuscript titled "A Behavioral Signature for Quantifying the Social Value of Interpersonal Relationships with Specific Others" has now been seen by 3 reviewers, whose comments are appended below. I have discussed the reports with my colleagues and I regret to inform you that we decided that in light of the referee reports, we cannot publish your manuscript in *Communications Psychology*.

You will see that the reviewers raise substantive concerns. Taking these points together with our editorial considerations, these reservations preclude publication of this study in *Communications Psychology*.

Editorially, we share the reviewers' interest and enthusiasm for the topics of your study, and we find your approach promising. However, we also share their concerns about your proposed measure's psychometric structure (Reviewers 1 and 3), its specificity and sensitivity (Reviewer 2), its relation to (superiority over?) competing measures, control and confound variables (Reviewer 1 and 3), as well as the validity of the underlying questionnaire (Reviewers 2 and 3).

We feel that those issues constitute a fundamental challenge to the validity of the measure, such that addressing them would require, at a minimum a complete new set of analyses, but could also necessitate a new item-scoring experiment e.g. eliciting judgments about enjoyment, rather than time spent for activities. In summary, although we find your study of potential interest, the significant concerns regarding the methodology necessitate an amount of additional data and analysis that we consider beyond the scope of what we can ask in revision.

Note, however, that we would be open to considering future work on this topic should it address the reviewers' methodological concerns.

I am sorry that we cannot be more positive on this occasion and thank you for the opportunity to consider your work.

Best regards,

Mael Lebreton

Mael Lebreton, PhD
Editorial Board Member
Communications Psychology
orcid.org/0000-0002-2071-4890

REVIEWERS' EXPERTISE:

Reviewer #1: valuation, social decision making

Reviewer #2: valuation, social decision making

Reviewer #3: groups, social decision making

REVIEWER COMMENTS:

Reviewer #1 (Remarks to the Author):

The authors proposed a novel measure of the value of social relationships, the social value score. The social value score operationalizes the value of social relationship into a weighted sum of likelihood rating of selected social behaviors. The authors also validated the social value score by examining its correlation with other social relationship related measures using two studies, one exploratory study and one pre-registered confirmatory study. They pointed out deviations from their pre-registration and made their data, code, and study materials publicly available on OSF.

Overall, I think this work could be a nice contribution to the field. Although the measure of social value score has potential in both its theoretical contribution and applied uses, there are a few issues that need to be addressed in a thorough revision. My concerns are detailed below, and I hope the authors can use them to improve their paper.

Major concerns

1) Unless I missed it, the proposed measure seems to be lacking some important assessments of its psychometric properties, notably dimensionality and measurement invariance of selected social activities. In essence, the social value score is a weighted sum of likelihood ratings of a series of selected activities to measure the value of social relationships. This means the factor underlying the activity items should be unidimensional. Dimensionality check is an important step in measurement development (e.g. Boateng et al., 2018). The assumed unidimensionality of the social value score should

be tested, is testable, and has been yet untested. Without such dimensionality evidence, it is difficult to interpret what the social value score actually measures. Along this line of measurement development, measurement invariance of the activity items used in the social value score is another important feature to consider and test (e.g. Putnick & Bornstein, 2016), such as if there is measurement invariance across age (teens and young adults such as those participants from UCLA SONA vs older adults from MTurk) and across types of relationship (parent vs friend vs acquaintance). Without evidence of measurement invariance, it is difficult to interpret the difference observed in social value scores.

2) The conclusion the authors' findings "show that social value is a unique motivator" and "is not equivalent to related constructs such as relationship quality" is not well supported. For a claim of uniqueness, I expected to see that the social value score is significantly associated with a DV after controlling for related constructs, such as relationship quality. In the analyses in the manuscript, I did not remember seeing any measure of "related constructs" that was included as a control variable in the analyses even though there are such measures in the data the authors collected, such as relationship quality, which was initially planned to be a control variable in the pre-registration. Although the authors explained in pre-registration deviation why they removed relationship quality as a control variable from the analyses, the argument is not convincing and there are better and more essential reasons to have relationship quality as a control variable in the analyses to show at least the uniqueness of social value score compared to relationship quality.

Minor concerns

1) I think the readability of the manuscript will be much improved if they authors move the Methods after the Introduction. As a method paper, presenting methods in such a scattered fashion makes it difficult for me to fully appreciate and evaluate the method. I imagine it will be similarly difficult for the general audience as well, especially those who want to apply the method to their own study.

2) The authors should say more about the feasibility of using this method. For example, the authors do not mention how much time it takes to finish the likelihood rating of their 56 or 70 selected activity items and retention rate. Without such important feasibility information, it is difficult to evaluate the practicality of a method regardless of its theoretical contribution.

3) Line 262-263, the authors mention they drastically increased the sample size due to large posterior variances observed in the exploratory analysis and used $N = 699$. How did

the authors exactly come up with this specific sample size? Being underpowered and overpowered can both be concerning (Cunningham & McCrum-Gardner, 2007; Bacchetti et al., 2005).

4) Line 388, “Prominent theories”. Some citations of the theories or a review of them are missing here.

5) Line 405-406, “proved to be related to...”, citation is needed.

6) Figure 2 and 3 caption, “89% highest density credible interval”, how was the threshold of “89%” determined?

Reviewer #2 (Remarks to the Author):

In this paper, the authors reported their methods for developing and validating a behavioral signature of social value using data across multiple independent samples. This signature is inspired by economic opportunity cost principles and data-driven quantitative techniques. Specifically, the authors formulated this signature by assessing and weighing a wide variety of social behaviors based on individuals' tendencies to prioritize them when resources are limited. The authors then evaluated how well this signature captured participants' self-reported social behaviors with specific relationship partners (such as a parent, a close friend, and an acquaintance). The “pattern expression” of this signature correlated with various aspects of various social relationships. Furthermore, these scores predicted decision-making preferences across a range of tasks involving the allocation of time, money, and other socio-affective resources. Overall, the paper addressed conceptually novel research questions and developed a useful tool for studying social behaviors. I have a couple of clarification questions for the authors (in no particular order).

1. The authors used time spent with a relationship partner as an index of the value an individual assigns to that relationship partner. Although I agree that this assumption is largely valid, I do think it is worth making the distinction between “wanting to” and “having to” spend time with a particular relationship partner. For example, for someone who is in a distant romantic relationship, they may only spend a few days with their romantic partner every month, but spend 5 days/week with their co-workers. That doesn't mean that this person values their co-workers. In other words, duration and enjoyment of the time spent

with a partner may not always go together. The authors acknowledge the inconsistency in the results regarding time spent with partners. I would suggest that the authors consider ways to integrate the quality and quantity aspects of the time spent measure (e.g., duration weighted by enjoyment).

2. If I understand the methods correctly, the authors asked the participants to evaluate the same set of activities in different relationship contexts. However, the same activity may have very different social and moral implications in different relationship contexts. As an extreme example, engaging in a sensual massage will be appropriate and enjoyable between romantic partners, but highly inappropriate (and less enjoyable) between parents and children (e.g., Earp et al., 2021 <https://www.nature.com/articles/s41467-021-26067-4>). Of course, this is an extreme example, and the activities the authors used in this study may not be as extreme as this one. But I encourage the authors to think more about the different social and moral implications of an activity in different relationship contexts.

3. For a brain-based signature, one typically assesses the signature's specificity and sensitivity (Chang et al., 2015; <https://journals.plos.org/plosbiology/article?id=10.1371/journal.pbio.1002180>). For specificity, one needs to test whether the predictive power of the said signature is specific to the construct in question (e.g., a brain-based signature of pain should have high predictive power for predicting different levels of pain, but not different levels of sadness). For sensitivity, one needs to assess to what extent the predictive power of the signature can be generalized to similar experiences (e.g., a brain-based signature of pain developed based on thermal pain should have decent predictive power when applied to electric shocks). How should we make sense of specificity and generalizability in the context of the behavioral signature the authors developed? Is there a way to evaluate their signature with regard to these values?

4. It is probably beyond the scope of the present research, but I would like to know a bit more about the authors' ideas about the generalizability of their behavioral signature across cultures. The relationships and activities tested in the present research are representative of the US (and perhaps other WEIRD) contexts. Will the signature hold when applied to relationships and activities in a different culture, where the valuation of relationships and activities is substantially different from the US? What would the consistencies and inconsistencies across cultures tell us about the nature of our valuation of social behaviors and relationships?

Reviewer #3 (Remarks to the Author):

In the present manuscript, the authors propose and evaluate a new method to quantify the value of social relationships. This method deviates from self-report measures employed in psychology, which often simply ask participants to directly evaluate their subjective valuation. Instead, it asks how likely people would spend their time with others across different activities, weighted in importance based on a separate task and sample. In their motivation, the authors draw inspiration from concepts important in economics, such as revealed preferences or opportunity costs.

Generally, I find the approach interesting. The general idea—that leisure time is scarce and deciding with whom people (hypothetically) spend this time can reveal something about the value they place on the relationship—is quite appealing and promising. The authors also employ advanced statistical methods, and the results are openly reported, although mostly in the Supplementary Information (SI). The paper is well-written, though the structure is somewhat challenging to follow (see below).

However, I do have some major concerns about the validity of the “Activity Weights,” underlying assumptions of the method (i.e., the degree to which reports reveal preferences) that are not completely spelled out, and the general value of the method compared to self-report measures. While I agree with the authors that “deriving a behavioral signature of interpersonal value” can be valuable and (possibly) more valid than self-reports that rely on the accuracy of introspection, the data did not completely convince me that this advantage is clearly documented empirically.

Below, I outline my concerns in more detail.

(1) The validity of derived “Activity Weights”

The authors asked an independent sample to weigh different activities, using this question as a proxy for how “preferred” certain activities are: “Suppose you had a couple of hours of time wherein you had no obligations or commitments. Which of the following activities are you most likely to do in this time? Which would you be least likely to do?”

From my understanding, activity weights are used as a proxy for how people perceive the value of different activities. However, I don't see to what degree this assumption is valid based on the question that was asked. It appears that the authors assume that, since free time is valuable, people will choose what they like most.

Another concept from economic decision models that may be valuable to introduce is the idea of costs. Assuming that different activities also have different costs attached, what people may do with their free time may not necessarily relate to how valuable they perceive the different activities. For example, according to Table S3, one of the highest weights is given to "browsing the internet" (2.6), whereas attending a musical event (-0.4) or sporting event (-0.6) has much lower weights. It makes sense to me that people would rank browsing the internet higher when asked "Which of the following activities are you most likely to do in this time?" because it is 'cheap' to do so. Attending a musical event is 'costlier,' and the opportunity also needs to arise; hence, people may see it as less likely. The important point is that I am not sure what the activity weight is proxying for. If it is supposed to reveal preferences, I am uncertain if that is confounded with the opportunity costs of different activities. I could imagine that attending a musical event is more preferred than browsing the internet when keeping costs constant (as the former is financially costly and the opportunity mostly arises in the evening, whereas the latter is cheap and easily available, etc.). In other words, what is measured here is simply what people do, not what they prefer to do. The assumption of revealed preferences may not hold, possibly due to the reasons mentioned (different costs/availability for different activities).

If I am not missing an important point here, the authors should (a) clearly state what the Activity Weights are supposed to represent and (b) explain why their elicitation method is a valid proxy for that. Furthermore, it would be valuable to examine what happens to the correlations with other measures (dictator game, social loss aversion, relationship quality, etc.) when all Activity Weights are set to 1 and compare it to the proposed model (akin to model comparison). This statistical approach would help determine whether the Activity Weights actually add explanatory power or simply introduce noise.

(2) Interrater reliability – How idiosyncratic are ratings?

Relatedly, from my understanding, Activity Weights were measured in a separate sample. This, to some degree, makes sense to me, as the approach is much more general if "general preference rankings" can be used to calculate "social value." Yet, this also assumes that there is some degree of agreement across raters, as mentioned briefly in the discussion. If preferences are completely idiosyncratic (some prefer A, some B, some C), this 'leap of faith' is not warranted. Since the authors asked a sample of participants to rank the same activities, it would be valuable to calculate some measure of interrater agreement to show that aggregating across participants and calculating one "Activity Weight" for each activity actually makes sense.

(3) The validity of "Likelihood Ratings"

I have a similar issue with the measure of Likelihood Ratings:

“Assume that you had a free day with no imminent obligations or commitments. Assume that this day is taking place during an ‘average’ month for you. How likely are you to engage in the following activities with the [PERSON] you nominated? In this scenario, assume that you live near your [PERSON] and that you could see them relatively easily if you wanted to.” The underlying assumption seems to be: If I prefer Person A over Person B, I am more likely (p) to engage in an activity that I personally enjoy more. Yet, it could also be that I specifically choose an activity that I do not like, but for which the other person adds value (i.e., due to some form of complementarity) – for example, doing my taxes because my parents are very good at it and can help me. For that reason, I may value the relationship a lot, even if the activity is not enjoyable (or becomes more enjoyable if done together). More generally, the underlying assumptions of the measures and the underlying model should be more clearly specified and critically assessed, respectively. At the moment, it seemed to me that a lot of hidden assumptions about what the measures represent and how they relate to each other are made, and some of these assumptions may not completely hold, threatening the validity of the “social value” measure. This could also explain the rather low correlations with other self-reports and measures, which are supposed to proxy for social relationship value.

(4) The factor structure of all measurements

The authors use many other measures to assess convergent validity. This leads to a lot of individual comparisons and associations (that are not corrected for multiple comparisons, as far as I understand). A possibly more stringent approach would be to, first, factor-analyze (or fit a structural equation model to) all the different measures (loss aversion, relationship quality, dictator game, forced choice, etc.). Do they all measure the same underlying construct (i.e., “relationship value”)? What is their factor structure? This may allow reducing all these measures to one (or two) construct(s) and correlating it with the results of the proposed social value measure. It could also explain why some correlations are low if different underlying constructs (i.e., low correlations between) are measured. In general, I am not an expert in psychometrics and the validation of psychological tests, but many methods from this literature (how to assess reliability, convergent and divergent validity, etc.) could or should be used here, as the authors seem to propose a new measure which, in my mind, should go beyond simply describing a new elicitation method and correlate it with (proposedly) similar measures.

(5) Interpretation of zero correlations

The authors say that “The goal of this study was to develop and validate a behavioral signature of social value that could be applied to quantify the value of interpersonal

relationships.” What I was left wondering: Is it now a more valid measure than the ‘competing’ measures? Is it better in predicting (but what exactly)? Or is it simply that the sometimes low or zero correlations with other measures is a sign that the new measure is noisy and lacks validity?

While I think the idea of developing a “behavioral signature” – asking for behavior instead of self-reported evaluations and exploiting the idea of opportunity costs – is really nice, my main concern, as also outlined in the point above, is that I was left questioning the validation of the measure and the added value.

(6) Information of the measures in the main-text

The writing of the paper was sometimes hard to follow. I understand that the structure of the journal requires the methods to be after the main text. Yet, the authors chose to provide a lot of statistical details also in the main text, while offering little information on the actual measurements and elicitation methods. It would really help to move some of the information on the measurements (what was exactly measured and how?) to the main text, as this is such a central part of the paper and is difficult to understand without consulting the methods (and SI), first.

(7) The distinction between friend and acquaintance already seems to imply a value judgment of different relationships. In that sense, it is a bit tautological. But this could actually be exploited more: I apologize if I missed that, but do people have lower likelihood ratings overall when paired with an acquaintance vs. a friend, and how large is this effect? This could serve as a simple and straightforward ‘manipulation check,’ increasing trust in the validity of this component of the measurement.

In sum, the authors, in my mind, need to show the (1) agreement in activity weights across raters (to warrant aggregation) and (2) additional explanatory power of adding activity weights in order to validate this part of the measurement (i.e., model comparison). Second, show that likelihood ratings already (strongly) distinguish friends from acquaintances. Third, provide a factor-structure of all additional measures used to validate the “behavioral signature” and show how the resulting latent factors correlate with the new measure. Lastly, clearly provide underlying assumptions related to revealed preferences and why the reader should believe that these assumptions are (at least partially) warranted. And lastly, provide reasons why (or when) people should use this new measure.

Author Responses: first round.

Reviewer 1

1) Unless I missed it, the proposed measure seems to be lacking some important assessments of its psychometric properties, notably dimensionality and measurement invariance of selected social activities. In essence, the social value score is a weighted sum of likelihood ratings of a series of selected activities to measure the value of social relationships. This means the factor underlying the activity items should be unidimensional. Dimensionality check is an important step in measurement development (e.g. Boateng et al., 2018). The assumed unidimensionality of the social value score should be tested, is testable, and has been yet untested. Without such dimensionality evidence, it is difficult to interpret what the social value score actually measures. Along this line of measurement development, measurement invariance of the activity items used in the social value score is another important feature to consider and test (e.g. Putnick & Bornstein, 2016), such as if there is measurement invariance across age (teens and young adults such as those participants from UCLA SONA vs older adults from MTurk) and across types of relationship (parent vs friend vs acquaintance). Without evidence of measurement invariance, it is difficult to interpret the difference observed in social value scores.

We agree with the reviewer that it is important to further probe the psychometric properties of the social value scores. One challenging aspect of running additional psychometric analyses is that many such analyses assume that composite scores from a given measure will be unit-weighted (i.e., all items weighted by 1) (e.g., Rodriguez, Reise, & Haviland, 2016, *Psych Methods*), which does not apply to the social value score method that we propose here. This does not indemnify our proposed method from evaluation by such tests, but it does require nuance in thinking about what their results mean when applied to ‘raw’ likelihood ratings versus the final social value scores.

First, we must consider the that unit-weighted likelihood scores and social value scores tap related but distinct constructs. Likelihood ratings capture how likely one is to do something with a particular known other in their free time (no commitments or obligations). An average of such scores describes how likely one would spend time with a given known other over a range of activities. Applying the behavioral signature weights reveals the extent to which one’s responses follows a pattern of idealized social behavior, indicating a different construct. It is analogous to calculating pattern expression values in cognitive neuroscience or polygenic risk scores in behavioral genetics. This is relevant because fitting factor analytic models to the likelihood ratings tells us more about the likelihood ratings than social value scores. The reliability of the likelihood ratings absolutely affects the reliability of the social value scores, but more must be done to verify the reliability and validity of social value scores. This is a lengthy preamble, but a worthwhile one in our view because it is important keep in mind there are multiple points of evaluation needed to better estimate the validity of social value scores.

To this end, we began by estimating the model-based reliability of the likelihood ratings using coefficient omega. Coefficient omega is a more appropriate test of reliability (Flora, 2020, *AMPPS*) because it relies on fewer assumptions than other methods (e.g., alpha). This procedure and results are detailed on Pg. 13:

“Internal Consistency. In order for social value scores to be useful, they must be reliable. One way to begin to evaluate the reliability of said scores is to determine the internal consistency of the likelihood ratings, a key substrate of social values scores. Before doing so, there are two related points to be kept in mind. First, unit-weighted (i.e., sum or mean) likelihood scores are not measuring the same construct as our social value scores. Second, and relatedly, measures of internal consistency typically evaluate the reliability of composite scores assuming unit weighting of instrument items. Because our social value scores are not derived from unit-weighted likelihood ratings, it is important to keep in mind that the reliability of the likelihood ratings are only part of the evaluation for the reliability of social value scores, and that relying solely on the application of classic psychometric techniques to likelihood ratings can lead to potentially misleading conclusions. However, ascertaining the reliability of likelihood ratings is still necessary for establishing the reliability of social value scores.

With this in mind, we first computed model-based reliability of likelihood ratings from the exploratory and confirmatory samples for both the parent, friend, and acquaintance ratings. We focused on model-based indices of reliability – ω_{total} , $\omega_{\text{hierarchical}}$, and explained common variance (ECV). These indices are computed from bi-factor models that decompose covariance between observed indicators as a function of a general latent factor—contributing to all indicators—and several group factors that contribute to a subset of items. ω_{total} captures the variance accounted for by all latent factors, $\omega_{\text{hierarchical}}$ captures the amount of total variance captured by the general factor, and the ECV captures how much of the common variance is accounted for by the general factor. The first index is a measure of the overall reliability of an instrument, whereas the latter two capture the general factor saturation. That is, some instruments may be multi-dimensional in theory, but may be unidimensional in practice due to having a strong common factor. We computed these indices using the omega function from the psych R package. For sensitivity, we varied the number of group factors (3, 4, 5). Likelihood ratings displayed internal consistency and strong general factor saturation both in the sample with SONA-sourced activities (Parent: ω_{total} 0.982 – 0.983, $\omega_{\text{hierarchical}}$ 0.759 – 0.883, ECV 0.592 – 0.741; Friend: ω_{total} 0.982 – 0.984, $\omega_{\text{hierarchical}}$ 0.732 – 0.786, ECV 0.567 – 0.635; Acquaintance: ω_{total} 0.987 – 0.989, $\omega_{\text{hierarchical}}$ 0.774 – 0.808, ECV 0.641 – 0.679) and in the sample with MTurk-sourced activities (Parent: ω_{total} 0.972 – 0.975, $\omega_{\text{hierarchical}}$ 0.651 – 0.700, ECV 0.493 – 0.552; Friend: ω_{total} 0.972 – 0.975, $\omega_{\text{hierarchical}}$ 0.681 – 0.708, ECV 0.484 – 0.552; Acquaintance: ω_{total} 0.984 – 0.985, $\omega_{\text{hierarchical}}$ 0.775 – 0.799, ECV 0.637 – 0.703). These results suggest excellent reliability and a practically unidimensional measure for all three types of relationships that were assessed here.”

As an aside, we note to the reviewer that unidimensionality is not a requisite condition for using single sum or mean scores from a measure (Flora, 2020, *AMPPS*). Indeed, there are plenty of instruments that are not unidimensional and can still be used to compute a sum score owing to the factor structure of the instrument. In our case, our data suggests that likelihood ratings are *essentially* or *practically* unidimensional (again see Rodriguez et al., 2016, *Psych Methods*) obviating any concerns that it is inappropriate to sum or average over a metric derived from the

likelihood ratings (even if it was not unidimensional, weighing each item by activity weights could potentially also obviate the concern, but this possibility need not be discussed further in light of the results reported above).

However, as we mentioned above, more is needed to ensure that social value scores—not just likelihood ratings—are reliable. This is why we went a step further and computed a ‘shuffle test’ (a permutation test) to verify whether the behavioral signature weights meaningfully contributed to the calculation of social value (Pg. 14):

“Confirming the Utility of Behavioral Signature Weights. To further verify the reliability of the social value scores, we checked to see whether the weights from the behavioral signature were meaningfully contributing to the calculation of social value scores. Said differently, we checked to make sure that the link between an activity and its weight was not arbitrary. To do this, we randomly shuffled weights of the behavioral signature, computed social value scores with the shuffled weights, and then re-ran several statistical analyses involving social value scores. We specifically re-ran correlations with relationship quality and social loss aversion, as well as paired differences between social value scores of each known other. Other analyses were not re-run in this test given computational demands.

Across both confirmatory datasets used here, we consistently observed that repeatedly shuffling the pairing between weights and likelihood ratings for a given activity and then re-computing standardized test statistics of interest resulted in distributions largely centered around a null value. This was true in the confirmatory sample with SONA-sourced activities for both for correlations between social value scores and relationship quality (Parent: $r_{\text{shuffled}} = -0.039$ (.14), Friend: $r_{\text{shuffled}} = -0.022$ (.13), Acquaintance: $r_{\text{shuffled}} = -0.040$ (.14)) and for correlations between social value scores and social loss aversion (Parent: $r_{\text{shuffled}} = -0.036$ (.14), Friend: $r_{\text{shuffled}} = -0.020$ (.10), Acquaintance: $r_{\text{shuffled}} = -0.054$ (.13)). This was also true in the confirmatory sample with SONA-sourced activities, both for correlations between social value scores and relationship quality (Parent: $r_{\text{shuffled}} = -0.098$ (.11), Friend: $r_{\text{shuffled}} = -0.069$ (.11), Acquaintance: $r_{\text{shuffled}} = -0.141$ (.14)) and for correlations between social value scores and social loss aversion (Parent: $r_{\text{shuffled}} = -0.085$ (.14), Friend: $r_{\text{shuffled}} = -0.069$ (.13), Acquaintance: $r_{\text{shuffled}} = -0.152$ (.13)). A similar pattern of null results was found when examining paired differences between social value scores for each known other using shuffled weights, both in the confirmatory sample with SONA-sourced activities (Parent – Friend: $d_{\text{shuffled}} = 0.033$ (SD = 0.37), Parent – Acquaintance: $d_{\text{shuffled}} = -0.018$ (0.39), Friend – Acquaintance: $d_{\text{shuffled}} = -0.056$ (0.26)) and in the confirmatory sample with MTurk-sourced activities (Parent – Friend: $d_{\text{shuffled}} = 0.087$ (0.33), Parent – Acquaintance: $d_{\text{shuffled}} = -0.090$ (0.41), Friend – Acquaintance: $d_{\text{shuffled}} = -0.185$ (0.27)). The one mild exception to the tendency for this approach to yield test statistics centered on a null value involved statistics related to acquaintance scores in the sample with MTurk-sourced weights, which were slightly elevated (around ~ 0.1 for d and r values) in the opposite direction. While we point this out in the interest of full disclosure, we are unconcerned with this finding because these effect sizes are still quite small and the variance of the shuffled distributions around them is still quite large.”

Finally, though not reported in the manuscript, we note that social value scores are highly correlated in the anticipated direction (inverse) when computing a scores from the subset of positive weights and another set of scores from the subset of negative weights (all correlations at

least -0.7). We are happy to insert these findings into the manuscript if the reviewer feels it will strengthen the quality of the paper.

In our view, these pieces of evidence help address psychometric concerns about the reliability of the social value scores in each of their constituent parts (likelihood ratings, behavioral signature weights). We would be happy to incorporate additional analyses, if needed, to address any remaining concerns in this vein.

2) The conclusion the authors' findings "show that social value is a unique motivator" and "is not equivalent to related constructs such as relationship quality" is not well supported. For a claim of uniqueness, I expected to see that the social value score is significantly associated with a DV after controlling for related constructs, such as relationship quality. In the analyses in the manuscript, I did not remember seeing any measure of "related constructs" that was included as a control variable in the analyses even though there are such measures in the data the authors collected, such as relationship quality, which was initially planned to be a control variable in the pre-registration. Although the authors explained in pre-registration deviation why they removed relationship quality as a control variable from the analyses, the argument is not convincing and there are better and more essential reasons to have relationship quality as a control variable in the analyses to show at least the uniqueness of social value score compared to relationship quality.

A subset of our post-hoc follow-up analyses in response to reviewer comments tested whether social value scores were still predictive of social behavior over and above relationship quality and social loss aversion, Pgs. 15-16:

"Are Social Value Scores Predictive of Behavior When Controlling for Other Facets of Relationships? Having further established the reliability and validity of social value scores, we next checked whether social value scores are predictive of behaviors when controlling for other facets of relationships. Given our available data, this meant regressing social choice preferences on the dictator game or the forced choice question about spending time with one of two social partners onto social value scores, relationship quality, and social loss aversion. Because each of the two dependent values requires participants to decide between a pair of known others (e.g., parent versus friend, friend versus acquaintance), we differenced the relationship facets and social value scores before entering them into the model to for better interpretability. To illustrate this, we ran one model in which we subtract relationship facets and social value scores for friend from parent and then entered the differences into a model predicting allocations between parent and friend on the dictator game. We enacted this procedure in the three aforementioned samples using Bayesian regression (all predictors were standardized and received standard normal priors).

Results from the models are depicted in Supplementary Figures 4 – 5. Overall, social value scores showed modest but consistent associations with outcomes and most of the posterior mass in these analyses usually fell around either side of zero. Some associations between social value and choice behavior controlling for other relationship facets were stronger, but these results were inconsistent. These analyses suggest that social value is a modest yet persistent predictor of social behavior after adjusting for other facets of social relationships."

The reviewer can thus see that social value scores are thus less predictive of the behavioral measures in question than other facets of relationships. Crucially, *in light of the additional post-hoc analyses further validating the signature*, we view this as an interesting substantive finding, rather than something that fully invalidates our proposed method for engineering a behavioral signature of social value. We have amended the discussion section to that effect, Pgs. 16-17:

“Indeed, our results suggest that interpersonal relationship value is related to, but ultimately distinct from, relationship quality, as well as several other ways of evaluating relationships. While our findings also show that social value is a motivator of social behavior, our post hoc analyses lead us to conclude that other ways of assessing relationships (social loss aversion, relationship quality) are stronger predictors of the behavioral measures used here. Given that our method for calculating social value appears to yield reliable and valid scores, we interpret this finding as helping delineate the facets of social relationships that likely shape social behavior. Our evidence suggests that social value has a small effect in shaping behavior, but is partially eclipsed by relationship quality and social loss aversion. These findings are non-trivial in our view because they help untangle the nomological network of associations between relationship facets and social behavior.”

3) I think the readability of the manuscript will be much improved if they authors move the Methods after the Introduction. As a method paper, presenting methods in such a scattered fashion makes it difficult for me to fully appreciate and evaluate the method. I imagine it will be similarly difficult for the general audience as well, especially those who want to apply the method to their own study.

We agree with the reviewer and have restructured the manuscript accordingly.

4) The authors should say more about the feasibility of using this method. For example, the authors do not mention how much time it takes to finish the likelihood rating of their 56 or 70 selected activity items and retention rate. Without such important feasibility information, it is difficult to evaluate the practicality of a method regardless of its theoretical contribution.

We appreciate the reviewer nudging us to consider practical implications of our proposed method. We have added the following to the discussion (pg. 18):

“With this in mind, we note that a strength of our method is the relative ease of implementation, especially after signature weights have been derived. The activity sourcing procedure requires relatively few participants since most activities volunteered by participants are easily collapsed into superordinate categories. The process of paring down activities was quickly accomplished and verified by members of the research team, and could perhaps be automated via large language models in the future. Further, whether it be implementing the MaxDiff design to obtain data to compute activity weights that comprise the signature or administering likelihood ratings of the activities themselves, research processes related to engineering or applying the signature can easily be implemented in relatively little time. Last, administering the likelihood ratings can be achieved quickly, as we estimate that it takes five minutes or less to complete said ratings per

target individual. Coupled with the fact that the same signature weights can be applied repeatedly to novel samples from a population, we believe that this ‘behavioral signature’ approach to calculating social value could mirror the utility of comparable approaches in cognitive neuroscience (40).”

5) Line 262-263, the authors mention they drastically increased the sample size due to large posterior variances observed in the exploratory analysis and used N = 699. How did the authors exactly come up with this specific sample size? Being underpowered and overpowered can both be concerning (Cunningham & McCrum-Gardner, 2007; Bacchetti et al., 2005).

We agree with reviewers that sample size planning is a critical part of any quantitative study, and we apologize for lack of clarity surrounding the sample sizes. The justification for doing so is listed on Pg. 8:

“We collected additional data to confirm and expand upon the results observed in the exploratory phase of the study. Given that some statistical associations observed in the previous phase were in the expected direction but had large posterior variances, we decided to drastically increase the sample size (N = 635 across four independent subsamples). Participants were recruited from MTurk and Prolific.”

We also clarify the sample sizes for each of the independent samples (we apologize for an oversight leading to a clerical error in reporting the final sample size), Pg. 9:

“We decided to analyze the MTurk data, according to our pre-registration plan, in addition to the Prolific data. We kept the screened and cleaned MTurk data so that these data could serve as an additional test of generalizability between online populations. Thus, the results reported in this confirmatory phase come from four subsamples, fully crossed between the participant pools from which participants were drawn to obtain lists of activities (Prolific, MTurk) and to validate social value scores (Prolific, MTurk). Notably, although concerns about fraudulent responses led to a final MTurk sample for validating social value scores that was smaller than planned (N = 181; N = 82 with SONA-sourced weights, N = 99 with MTurk-sourced weights), by conducting additional recruitment via Prolific, we obtained a final sample of N = 454 (N = 233 with SONA sourced weights, N = 221 with MTurk sourced weights), consistent with our pre-registered data collection plan.”

N = 635 reflects the number of participants in this entire phase of the study. However, as we more clearly note above, we analyzed the data sequentially in four separate samples (two from MTurk, two from Prolific; two with SONA-sourced activities, two with MTurk-sourced activities). Power analyses for Bayesian methods are less straightforward than for frequentist methods, and to some extent it matters less – if a study is ‘overpowered’, the likelihood will dominate the posterior; if it is underpowered, the prior will dominate. In our case, we have set up the analysis so that priors are weakly informative around a null value to be conservative and

regularize our observed parameters. Moreover, we use the Region of Practical Equivalence (ROPE, see Pg. 8) method to help contextualize and interpret meaningful effect sizes. Thus, even if we are overpowered (unlikely for regression analyses with ~230 participants, compared to when similar concerns are voiced for large scale studies like ABCD or UK Biobank, which have tens of thousands of subjects), we are still focusing on *effect sizes* and the precision thereof.

6) Line 388, “Prominent theories”. Some citations of the theories or a review of them are missing here.

The passage referenced by the reviewer now reads as follows (Pg. 16):

“These results complement the vast literature on value-based human behavior. Prominent theories posit that human behavior is guided by cognitive and affective heuristics that compute the subjective value of prospective actions, and the action that maximizes subjective value is then enacted (50–54).”

7) Line 405-406, “proved to be related to...”, citation is needed.

We were referencing our own findings in this line and thus do not have a citation available from the prior literature. We have adjusted language to make this clearer for readers (Pg. 16):

“Indeed, our results suggest that interpersonal relationship value is related to, but ultimately distinct from, relationship quality, as well as several other ways of evaluating relationships.”

8) Figure 2 and 3 caption, “89% highest density credible interval”, how was the threshold of “89%” determined?

We now include our justification for using 89% credible intervals (Pg. 8): “We used 89% credible intervals upon the recommendation that wider intervals (e.g., 95%) are more sensitive to Monte Carlo sampling error (41,42).”

Reviewer 2

1. The authors used time spent with a relationship partner as an index of the value an individual assigns to that relationship partner. Although I agree that this assumption is largely valid, I do think it is worth making the distinction between “wanting to” and “having to” spend time with a particular relationship partner. For example, for someone who is in a distant romantic relationship, they may only spend a few days with their romantic partner every month, but spend 5 days/week with their co-workers. That doesn’t mean that this person values their co-workers. In other words, duration and enjoyment of the time spent with a partner may not always go together. The authors acknowledge the inconsistency in the results regarding time spent with partners. I would suggest that the authors consider ways to integrate

the quality and quantity aspects of the time spent measure (e.g., duration weighted by enjoyment).

We agree with the reviewer that this would likely heighten the effectiveness of our proposed method. However, we do not have data to currently speak to this notion. We now consider this issue (alongside other important extensions to be pursued in future research) in the discussion section, Pg. 18:

“We also envision several important extensions to be pursued in future studies, such as modulating likelihood ratings by anticipated or past enjoyment of completing a given activity with a particular social partner, incorporating information about how being accompanied to unenjoyable activities by certain social partners can heighten the experience, personalizing the activity weights, fine-tuning them to better reflect relationship by activity interactions, and incorporating the effort cost of each activity in the signature.”

We also note that some of the reviewer’s concern may be partly obviated by the prompt used to source likelihood activities:

“Assume that you had a free day with no imminent obligations or commitments. Assume that this day is taking place during an ‘average’ month for you. How likely are you to engage in the following activities with the [PERSON] you nominated? In this scenario, assume that you live near your [PERSON] and that you could see them relatively easily if you wanted to?”

As noted above, the prompt for the likelihood ratings above is meant to help mitigate against any potential duration-enjoyment ‘disequilibrium’. Though we acknowledge addressing this in a more direct and comprehensive manner is preferable, we do think this likely mitigates some of this concern. We apologize that this was not made sufficiently clear in the original manuscript, and we hope that the revised manuscript structure will help to clarify this for future readers.

2. If I understand the methods correctly, the authors asked the participants to evaluate the same set of activities in different relationship contexts. However, the same activity may have very different social and moral implications in different relationship contexts. As an extreme example, engaging in a sensual massage will be appropriate and enjoyable between romantic partners, but highly inappropriate (and less enjoyable) between parents and children (e.g., Earp et al., 2021 <https://www.nature.com/articles/s41467-021-26067-4>). Of course, this is an extreme example, and the activities the authors used in this study may not be as extreme as this one. But I encourage the authors to think more about the different social and moral implications of an activity in different relationship contexts.

We appreciate this comment by the reviewer. The idea that certain activities vary in their appropriateness based on relationship type did not escape us, and we previously listed two measures we enacted in an attempt to make the signature generalizable across different relationship types.

First, we took the raw activities listed by participants and shifted them into a slightly higher level of abstraction. This should help make the activities slightly more generalizable (e.g., ‘watching a movie’ is reasonably applicable across parents, friends, and acquaintances, whereas watching, say, *Saltburn*, a movie with a particularly controversial bathtub scene, might be less appealing to do with a parent or an acquaintance). We have revised the relevant portion of the Methods section to make this clearer (Pg. 5): “Describing activities at a slightly higher level of abstraction facilitated data aggregation across respondents and their social partners, and in so doing, created a behavioral signature of social value that could generalize across social partners and respondents.”

Second, relationship-activity boundaries are likely to vary across individuals, possibly influencing one’s prioritization of a given activity (e.g., perhaps activities that are highly generalizable across relationship categories to a given individual are more likely to be pursued in their free time) and thus affecting their ratings on the Max Diff procedure for deriving behavioral signature weights. To safeguard against this, we scaled the model’s activity weights by the random subject variance (Pg. 32): “Posterior means of the ‘fixed effects’ were scaled by the posterior variances of the ‘random effects’ to account for individual differences in preferences among subjects.” Activities that are then highly variable in their prioritization across subjects had their weights adjusted closer to zero in a way that was proportional to the degree of between subject variability (this procedure was not enacted to mitigate against just this concern, as it was implemented to make the behavioral signature broadly resilient to individual differences, but we feel it applies here in this case; the reviewer can see how scaling affects the magnitude of the weights in Supplementary Tables 2-3).

Last, we are happy to repeat analyses without the inclusion of any particular activities that the reviewer thinks are not readily generalizable across different relationship categories. All activities (and their corresponding weights) are listed in the supplement.

3. For a brain-based signature, one typically assesses the signature’s specificity and sensitivity (Chang et al., 2015; <https://journals.plos.org/plosbiology/article?id=10.1371/journal.pbio.1002180>). For specificity, one needs to test whether the predictive power of the said signature is specific to the construct in question (e.g., a brain-based signature of pain should have high predictive power for predicting different levels of pain, but not different levels of sadness). For sensitivity, one needs to assess to what extent the predictive power of the signature can be generalized to similar experiences (e.g., a brain-based signature of pain developed based on thermal pain should have decent predictive power when applied to electric shocks). How should we make sense of specificity and generalizability in the context of the behavioral signature the authors developed? Is there a way to evaluate their signature with regard to these values?

We agree that this is a critical step for validating the signature that we were previously missing. As one of our *post hoc* analyses, we conducted tests of discriminant validity (Pg. 15):

“Social Value Scores Show Good Discriminant Validity. We next tested whether social value scores showed good discriminant validity (i.e., they were not correlated with theoretically unrelated constructs). To do this, we correlated social value scores for a given social partner with relationship quality and social loss aversion for all other partners. This test operates under the assumption that social value scores estimated for a given relationship (e.g., one’s acquaintance) should not be substantially predictive of relationship features (e.g., relationship quality, social loss aversion) of one’s other relationships. This meant that we ran 12 tests in each of the three samples (3 partners’ social value scores x 2 mismatched social partners x 2 outcome variables). To contextualize the correlation coefficients and judge discriminant validity, we adopted an empirical effect size heuristic threshold of $r = 0.12$ recently derived from the social psychology literature (49).

Overall, social value scores evinced good discriminant validity. Ten of the twelve correlations in the confirmatory sample with SONA-sourced activities resulted in correlation coefficients at or below 0.12. The two correlations exceeding the were parent social value score associations with friend relationship quality ($r = 0.309$) and acquaintance relationship quality ($r = 0.198$). All twelve correlations in the confirmatory sample with MTurk-sourced activities were estimated to be less than or equal to 0.12 and thus none reached the threshold for a medium effect size.

Following up on the correlations that exceeded the threshold, it occurred to us that those correlations may be driven by individual differences in how individuals place social value on relationships more generally. We reasoned one way to address this would be to re-compute the correlation between each pair of variables after partialing out the variance of social value scores belonging to the second person (e.g., parent social value scores were correlated with friend relationship quality after partialing out variance from friend social value scores). When doing so, the value of both correlations dropped substantially ($r_{\text{partial}} [\text{parent social value, friend relationship quality}] : 0.198$, $r_{\text{partial}} [\text{parent social value, acquaintance relationship quality}] : 0.077$). This suggests some of the covariance in these associations is attributable to non-relevant individual differences.”

We interpret the results as our method yielding social value scores that evince good discriminant validity.

4. It is probably beyond the scope of the present research, but I would like to know a bit more about the authors’ ideas about the generalizability of their behavioral signature across cultures. The relationships and activities tested in the present research are representative of the US (and perhaps other WEIRD) contexts. Will the signature hold when applied to relationships and activities in a different culture, where the valuation of relationships and activities is substantially different from the US? What would the consistencies and inconsistencies across cultures tell us about the nature of our valuation of social behaviors and relationships?

We appreciate the reviewer bringing this up and now consider the issue in the discussion (Pg. 17-18):

“Particular care will need to be taken when implementing this method in different cultures. While we currently believe that the method, though not necessarily the signatures derived here, are likely to generalize to other WEIRD samples (Western, Educated, Industrialized, Rich, Democratic; (65)), this may not be the case for non-WEIRD cultures. Future work seeking to implement this method should be aware of cultural dynamics that may complicate the applicability of the procedure proposed here. In particular, we could foresee a case where cultural norms around social activities in other countries more strongly dictate the acceptability of completing an activity with specific social partners, limiting the degree to which participants could willfully choose how to spend their time in social contexts. In such a case, new items that meaningfully track with the expenditure of finite social capital would need to be identified and used instead of activities completed in one’s leisure time.”

Reviewer 3

(1) The validity of derived “Activity Weights”

The authors asked an independent sample to weigh different activities, using this question as a proxy for how "preferred" certain activities are: "Suppose you had a couple of hours of time wherein you had no obligations or commitments. Which of the following activities are you most likely to do in this time? Which would you be least likely to do?"

From my understanding, activity weights are used as a proxy for how people perceive the value of different activities. However, I don't see to what degree this assumption is valid based on the question that was asked. It appears that the authors assume that, since free time is valuable, people will choose what they like most.

Another concept from economic decision models that may be valuable to introduce is the idea of costs. Assuming that different activities also have different costs attached, what people may do with their free time may not necessarily relate to how valuable they perceive the different activities. For example, according to Table S3, one of the highest weights is given to "browsing the internet" (2.6), whereas attending a musical event (-0.4) or sporting event (-0.6) has much lower weights. It makes sense to me that people would rank browsing the internet higher when asked "Which of the following activities are you most likely to do in this time?" because it is 'cheap' to do so. Attending a musical event is 'costlier,' and the opportunity also needs to arise; hence, people may see it as less likely.

The important point is that I am not sure what the activity weight is proxying for. If it is supposed to reveal preferences, I am uncertain if that is confounded with the opportunity costs of different activities. I could imagine that attending a musical event is more preferred than browsing the internet when keeping costs constant (as the former is financially costly and the

opportunity mostly arises in the evening, whereas the latter is cheap and easily available, etc.). In other words, what is measured here is simply what people do, not what they prefer to do. The assumption of revealed preferences may not hold, possibly due to the reasons mentioned (different costs/availability for different activities).

If I am not missing an important point here, the authors should (a) clearly state what the Activity Weights are supposed to represent and (b) explain why their elicitation method is a valid proxy for that. Furthermore, it would be valuable to examine what happens to the correlations with other measures (dictator game, social loss aversion, relationship quality, etc.) when all Activity Weights are set to 1 and compare it to the proposed model (akin to model comparison). This statistical approach would help determine whether the Activity Weights actually add explanatory power or simply introduce noise.

We appreciate the reviewer's thoughtful comment. We have taken several measures to address this comment. First, we take up their suggestion to more clearly state what the activity weight is proxying for (Pg. 4):

“The entire process of engineering the behavioral signature involved sourcing activities that would comprise the signature, deriving weights that quantified the extent to which individuals are willing to prioritize engagement in a particular activity in the face of opportunity cost, collecting data on how likely individuals are to complete each activity with specific social partners, and computing the expression of the behavioral signature in the aforementioned likelihoods. Each weight that comprises the signature reflects the rank order likelihood of prioritizing a given activity relative to all others; collectively they reflect a zero hierarchy of preferences for each activity. Subsequently, our logic is that this behavioral signature of social value is intended to represent an idealized allocation of finite leisure time across a variety of possible activities. We argue the extent to which one's activities with a given social partner conforms to this pattern should be reflective, to a meaningful degree, of the value one places on said partner (albeit not perfectly so).”

Some of the text above was included in the previous draft of the manuscript, but upon re-reading it we agree with the reviewer that more was needed.

As an aside, we acknowledge that the prompt could possibly be revised (though the reviewer is unclear about what changes they feel would enhance the face validity of the prompt) but we must note that a key component of the behavioral signature engineering process involves the MaxDiff design. The forced choice element of the MaxDiff design helps incorporate opportunity cost by adding a zero-sum quality to the weights.

Second, we refer the reviewer to the newly added ‘*Post Hoc Analyses: Further Psychometric Assessment of Social Value Scores*’ section (pg. xx) and our responses to Comment 1 by Reviewer 1 and Comments 2 by Reviewer 2. The newly added section includes several analyses

(including comparing value scores with unit weighted likelihood scores, as suggested by the reviewer) that show the activity weights are not simply a source of noise.

Finally, we note the importance of incorporating effort cost for the future studies (Pg. 18, emphasis added here):

“We also envision several important extensions to be pursued in future studies, such as modulating likelihood ratings by anticipated or past enjoyment of completing a given activity with a particular social partner, incorporating information about how being accompanied to unenjoyable activities by certain social partners can heighten the experience, personalizing the activity weights, fine-tuning them to better reflect relationship by activity interactions, and incorporating the effort cost of each activity in the signature.”

(2) Interrater reliability – How idiosyncratic are ratings?

Relatedly, from my understanding, Activity Weights were measured in a separate sample. This, to some degree, makes sense to me, as the approach is much more general if "general preference rankings" can be used to calculate "social value." Yet, this also assumes that there is some degree of agreement across raters, as mentioned briefly in the discussion. If preferences are completely idiosyncratic (some prefer A, some B, some C), this 'leap of faith' is not warranted. Since the authors asked a sample of participants to rank the same activities, it would be valuable to calculate some measure of interrater agreement to show that aggregating across participants and calculating one "Activity Weight" for each activity actually makes sense.

We agree that some form of interrater reliability would be helpful in order to ascertain the reliability of the activity weights themselves. While the reviewer suggests an elegant solution (thanks!), it is unfortunately not as straightforward as it seems due to the random ordering and subsetting of the MaxDiff design.

The idea that between-person idiosyncrasies could harm the validity of the weights did not previously escape our attention, as we scale each activity weight by the random effect variance from the hierarchical model applied to the MaxDiff data used to derive the weights. In other words, the weights are obtained by taking the ‘fixed effects’ from the model and then scaling them by the random effect variance. This latter quantity represents the degree of between-person heterogeneity for each weight. These variances are listed in Supplementary Tables 2 and 3.

The effect of scaling by this variance quantity is that the overall signatures should be less influenced by highly variable items, proportional to the magnitude of between-person differences. This reflects the nuanced reality of the problem identified by the reviewer – the question is not ‘is there between-person variability in the weights?’, it is ‘how much?’ seeing as no two people are alike.

Moreover, the hierarchical model itself also helps safeguard against this since activities that are ‘lower in agreement’ between subjects will naturally be assigned a lower weight to begin with.

In conjunction with our other analyses showing that the weights are not simply a source of noise (referenced above), we argue that the weights herein are psychometrically sound.

(3) The validity of “Likelihood Ratings”

I have a similar issue with the measure of Likelihood Ratings:

“Assume that you had a free day with no imminent obligations or commitments. Assume that this day is taking place during an ‘average’ month for you. How likely are you to engage in the following activities with the [PERSON] you nominated? In this scenario, assume that you live near your [PERSON] and that you could see them relatively easily if you wanted to.”

The underlying assumption seems to be: If I prefer Person A over Person B, I am more likely (p) to engage in an activity that I personally enjoy more. Yet, it could also be that I specifically choose an activity that I do not like, but for which the other person adds value (i.e., due to some form of complementarity) – for example, doing my taxes because my parents are very good at it and can help me. For that reason, I may value the relationship a lot, even if the activity is not enjoyable (or becomes more enjoyable if done together).

More generally, the underlying assumptions of the measures and the underlying model should be more clearly specified and critically assessed, respectively. At the moment, it seemed to me that a lot of hidden assumptions about what the measures represent and how they relate to each other are made, and some of these assumptions may not completely hold, threatening the validity of the “social value” measure. This could also explain the rather low correlations with other self-reports and measures, which are supposed to proxy for social relationship value.

We appreciate this comment by the reviewer. We first refer the reviewer to the ‘*Post Hoc Analyses: Further Psychometric Assessment of Social Value Scores*’ section for additional analyses that address various underlying assumptions behind our method. This section includes analyses showing high reliability of likelihood ratings (‘*Internal Consistency*’), confirming that behavioral signature weights are not simply added noise (‘*Confirming the Utility of Behavioral Signature Weights*’), showing that social value scores are predictive over and above unit weighted likelihood scores (‘*Social Value Scores Are Still Largely Related to Behavior when Controlling for Unit-Weighted Likelihood Ratings*’), and demonstrating that social value scores evince good discriminant validity (‘*Social Value Scores Show Good Discriminant Validity*’).

Second, we have noted future avenues for fine-tuning the method in the discussion in ways suggested by all reviewers (Pg. 18; this was quoted above but we reprint here for convenience):

“We also envision several important extensions to be pursued in future studies, such as modulating likelihood ratings by anticipated or past enjoyment of completing a given activity

with a particular social partner, incorporating information about how being accompanied to unenjoyable activities by certain social partners can heighten the experience, personalizing the activity weights, fine-tuning them to better reflect relationship by activity interactions, and incorporating the effort cost of each activity in the signature.”

Finally, we respectfully disagree with the reviewer’s judgment that zero-order correlations between social value scores and other measures are “rather low”. Several of the correlations involving social value scores exceed empirically established thresholds for medium effect sizes in social psychology (see Lovakov & Agadullina, 2021, *Eur Jour of Soc Psych*), and it is generally unrealistic to expect large effect sizes in behavioral sciences research except for differences between extreme conditions (e.g., pain before/after a cold presser task).

(4) The factor structure of all measurements

The authors use many other measures to assess convergent validity. This leads to a lot of individual comparisons and associations (that are not corrected for multiple comparisons, as far as I understand). A possibly more stringent approach would be to, first, factor-analyze (or fit a structural equation model to) all the different measures (loss aversion, relationship quality, dictator game, forced choice, etc.). Do they all measure the same underlying construct (i.e., “relationship value”)? What is their factor structure? This may allow reducing all these measures to one (or two) construct(s) and correlating it with the results of the proposed social value measure. It could also explain why some correlations are low if different underlying constructs (i.e., low correlations between) are measured.

In general, I am not an expert in psychometrics and the validation of psychological tests, but many methods from this literature (how to assess reliability, convergent and divergent validity, etc.) could or should be used here, as the authors seem to propose a new measure which, in my mind, should go beyond simply describing a new elicitation method and correlate it with (proposedly) similar measures.

This comment converges with Comment 1 by Reviewer 1 and Comment 3 by Reviewer 2. We reproduce our responses to those comments here for this reviewer’s convenience.

Response to C1.R1: “We agree with the reviewer that it is important to further probe the psychometric properties of the social value scores. One challenging aspect of running additional psychometric analyses is that many such analyses assume that composite scores from a given measure will be unit-weighted (i.e., all items weighted by 1) (e.g., Rodriguez, Reise, & Haviland, 2016, *Psych Methods*), which does not apply to the social value score method that we propose here. This does not indemnify our proposed method from evaluation by such tests, but it does require nuance in thinking about what their results mean when applied to ‘raw’ likelihood ratings versus the final social value scores.

First, we must consider the that unit-weighted likelihood scores and social value scores tap related but distinct constructs. Likelihood ratings capture how likely one is to do something with

a particular known other in their free time (no commitments or obligations). An average of such scores describes how likely one would spend time with a given known other over a range of activities. Applying the behavioral signature weights reveals the extent to which one's responses follows a pattern of idealized social behavior, indicating a different construct. It is analogous to calculating pattern expression values in cognitive neuroscience or polygenic risk scores in behavioral genetics. This is relevant because fitting factor analytic models to the likelihood ratings tells us more about the likelihood ratings than social value scores. The reliability of the likelihood ratings absolutely affects the reliability of the social value scores, but more must be done to verify the reliability and validity of social value scores. This is a lengthy preamble, but a worthwhile one in our view because it is important keep in mind there are multiple points of evaluation needed to better estimate the validity of social value scores.

To this end, we began by estimating the model-based reliability of the likelihood ratings using coefficient omega. Coefficient omega is a more appropriate test of reliability (Flora, 2020, AMPPS) because it relies on fewer assumptions than other methods (e.g., alpha). This procedure and results are detailed on Pg. 13:

“Internal Consistency. In order for social value scores to be useful, they must be reliable. One way to begin to evaluate the reliability of said scores is to determine the internal consistency of the likelihood ratings, a key substrate of social values scores. Before doing so, there are two related points to be kept in mind. First, unit-weighted (i.e., sum or mean) likelihood scores are not measuring the same construct as our social value scores. Second, and relatedly, measures of internal consistency typically evaluate the reliability of composite scores assuming unit weighting of instrument items. Because our social value scores are not derived from unit-weighted likelihood ratings, it is important to keep in mind that the reliability of the likelihood ratings are only part of the evaluation for the reliability of social value scores, and that relying solely on the application of classic psychometric techniques to likelihood ratings can lead to potentially misleading conclusions. However, ascertaining the reliability of likelihood ratings is still necessary for establishing the reliability of social value scores.

With this in mind, we first computed model-based reliability of likelihood ratings from the exploratory and confirmatory samples for both the parent, friend, and acquaintance ratings. We focused on model-based indices of reliability – ω_{total} , $\omega_{hierarchical}$, and explained common variance (ECV). These indices are computed from bi-factor models that decompose covariance between observed indicators as a function of a general latent factor—contributing to all indicators—and several group factors that contribute to a subset of items. ω_{total} captures the variance accounted for by all latent factors, $\omega_{hierarchical}$ captures the amount of total variance captured by the general factor, and the ECV captures how much of the common variance is accounted for by the general factor. The first index is a measure of the overall reliability of an instrument, whereas the latter two capture the general factor saturation. That is, some instruments may be multi-dimensional in theory, but may be unidimensional in practice due to having a strong common factor. We computed these indices using the omega function from the psych R package. For sensitivity, we varied the number of group factors (3, 4, 5). Likelihood ratings displayed internal consistency and strong general factor saturation both in the sample with SONA-sourced activities (Parent: ω_{total} 0.982 – 0.983, $\omega_{hierarchical}$ 0.759 – 0.883, ECV 0.592 – 0.741; Friend: ω_{total} 0.982 – 0.984, $\omega_{hierarchical}$ 0.732 – 0.786, ECV 0.567 – 0.635; Acquaintance: ω_{total} 0.987 – 0.989, $\omega_{hierarchical}$ 0.774 – 0.808, ECV 0.641 – 0.679) and in the

sample with MTurk-sourced activities (Parent: ω_{total} 0.972 – 0.975, $\omega_{hierarchical}$ 0.651 – 0.700, ECV 0.493 – 0.552; Friend: ω_{total} 0.972 – 0.975, $\omega_{hierarchical}$ 0.681 – 0.708, ECV 0.484 – 0.552; Acquaintance: ω_{total} 0.984 – 0.985, $\omega_{hierarchical}$ 0.775 – 0.799, ECV 0.637 – 0.703). These results suggest excellent reliability and a practically unidimensional measure for all three types of relationships that were assessed here.”

As an aside, we note to the reviewer that unidimensionality is not a requisite condition for using single sum or mean scores from a measure (Flora, 2020, AMPPS). Indeed, there are plenty of instruments that are not unidimensional and can still be used to compute a sum score owing to the factor structure of the instrument. In our case, our data suggests that likelihood ratings are essentially or practically unidimensional (again see Rodriguez et al., 2016, Psych Methods) obviating any concerns that it is inappropriate to sum or average over a metric derived from the likelihood ratings (even if it was not unidimensional, weighing each item by activity weights could potentially also obviate the concern, but this possibility need not be discussed further in light of the results reported above).

However, as we mentioned above, more is needed to ensure that social value scores—not just likelihood ratings—are reliable. This is why we went a step further and computed a ‘shuffle test’ (a permutation test) to verify whether the behavioral signature weights meaningfully contributed to the calculation of social value (Pg. 14):

“Confirming the Utility of Behavioral Signature Weights. To further verify the reliability of the social value scores, we checked to see whether the weights from the behavioral signature were meaningfully contributing to the calculation of social value scores. Said differently, we checked to make sure that the link between an activity and its weight was not arbitrary. To do this, we randomly shuffled weights of the behavioral signature, computed social value scores with the shuffled weights, and then re-ran several statistical analyses involving social value scores. We specifically re-ran correlations with relationship quality and social loss aversion, as well as paired differences between social value scores of each known other. Other analyses were not re-run in this test given computational demands.

Across both confirmatory datasets used here, we consistently observed that repeatedly shuffling the pairing between weights and likelihood ratings for a given activity and then re-computing standardized test statistics of interest resulted in distributions largely centered around a null value. This was true in the confirmatory sample with SONA-sourced activities for both for correlations between social value scores and relationship quality (Parent: $r_{shuffled} = -0.039$ (.14), Friend: $r_{shuffled} = -0.022$ (.13), Acquaintance: $r_{shuffled} = -0.040$ (.14)) and for correlations between social value scores and social loss aversion (Parent: $r_{shuffled} = -0.036$ (.14), Friend: $r_{shuffled} = -0.020$ (.10), Acquaintance: $r_{shuffled} = -0.054$ (.13)). This was also true in the confirmatory sample with SONA-sourced activities, both for correlations between social value scores and relationship quality (Parent: $r_{shuffled} = -0.098$ (.11), Friend: $r_{shuffled} = -0.069$ (.11), Acquaintance: $r_{shuffled} = -0.141$ (.14)) and for correlations between social value scores and social loss aversion (Parent: $r_{shuffled} = -0.085$ (.14), Friend: $r_{shuffled} = -0.069$ (.13), Acquaintance: $r_{shuffled} = -0.152$ (.13)). A similar pattern of null results was found when examining paired differences between social value scores for each known other using shuffled weights, both in the confirmatory sample with SONA-sourced activities (Parent – Friend: $d_{shuffled} = 0.033$ (SD = 0.37), Parent – Acquaintance: $d_{shuffled} = -0.018$ (0.39), Friend –

Acquaintance: $d_{\text{shuffled}} = -0.056$ (0.26)) and in the confirmatory sample with MTurk-sourced activities (Parent – Friend: $d_{\text{shuffled}} = 0.087$ (0.33), Parent – Acquaintance: $d_{\text{shuffled}} = -0.090$ (0.41), Friend – Acquaintance: $d_{\text{shuffled}} = -0.185$ (0.27)). The one mild exception to the tendency for this approach to yield test statistics centered on a null value involved statistics related to acquaintance scores in the sample with MTurk-sourced weights, which were slightly elevated (around ~ 0.1 for d and r values) in the opposite direction. While we point this out in the interest of full disclosure, we are unconcerned with this finding because these effect sizes are still quite small and the variance of the shuffled distributions around them is still quite large.”

Finally, though not reported in the manuscript, we note that social value scores are highly correlated in the anticipated direction (inverse) when computing a scores from the subset of positive weights and another set of scores from the subset of negative weights (all correlations at least -0.7). We are happy to insert these findings into the manuscript if the reviewer feels it will strengthen the quality of the paper.

In our view, these pieces of evidence help address psychometric concerns about the reliability of the social value scores in each of their constituent parts (likelihood ratings, behavioral signature weights). We would be happy to incorporate additional analyses, if needed, to address any remaining concerns in this vein.”

Response to C3.R2: *“We agree that this is a critical step for validating the signature that we were previously missing. As one of our post hoc analyses, we conducted tests of discriminant validity (Pg. 15):*

“Social Value Scores Show Good Discriminant Validity. We next tested whether social value scores showed good discriminant validity (i.e., they were not correlated with theoretically unrelated constructs). To do this, we correlated social value scores for a given social partner with relationship quality and social loss aversion for all other partners. This test operates under the assumption that social value scores estimated for a given relationship (e.g., one’s acquaintance) should not be substantially predictive of relationship features (e.g., relationship quality, social loss aversion) of one’s other relationships. This meant that we ran 12 tests in each of the three samples (3 partners’ social value scores \times 2 mismatched social partners \times 2 outcome variables). To contextualize the correlation coefficients and judge discriminant validity, we adopted an empirical effect size heuristic threshold of $r = 0.12$ recently derived from the social psychology literature (49).

Overall, social value scores evinced good discriminant validity. Ten of the twelve correlations in the confirmatory sample with SONA-sourced activities resulted in correlation coefficients at or below 0.12. The two correlations exceeding the were parent social value score associations with friend relationship quality ($r = 0.309$) and acquaintance relationship quality ($r = 0.198$). All twelve correlations in the confirmatory sample with MTurk-sourced activities were estimated to be less than or equal to 0.12 and thus none reached the threshold for a medium effect size.

Following up on the correlations that exceeded the threshold, it occurred to us that those correlations may be driven by individual differences in how individuals place social value on

relationships more generally. We reasoned one way to address this would be to re-compute the correlation between each pair of variables after partialing out the variance of social value scores belonging to the second person (e.g., parent social value scores were correlated with friend relationship quality after partialing out variance from friend social value scores). When doing so, the value of both correlations dropped substantially (r_{partial} [parent social value, friend relationship quality]: 0.198, r_{partial} [parent social value, acquaintance relationship quality]: 0.077). This suggests some of the covariance in these associations is attributable to non-relevant individual differences.”

We interpret the results as our method yielding social value scores that evince good discriminant validity.”

In sum, social value scores seem to tap a practically unidimensional measure that is reliable (both in its likelihood ratings and weights) and valid (it is correlated with things it is related to, as previously shown; it is not correlated with things it should not be related to; it is predictive of behavior over and above unit-weighted likelihood ratings).

We also highlight that we have conducted various other psychometric tests in response to all reviewers’ comments. As we note above, the ‘*Post Hoc* Analyses: Further Psychometric Assessment of Social Value Scores’ section contains many additional analyses that address various underlying assumptions behind our method.

(5) Interpretation of zero correlations

The authors say that “The goal of this study was to develop and validate a behavioral signature of social value that could be applied to quantify the value of interpersonal relationships.” What I was left wondering: Is it now a more valid measure than the ‘competing’ measures? Is it better in predicting (but what exactly)? Or is it simply that the sometimes low or zero correlations with other measures is a sign that the new measure is noisy and lacks validity?

While I think the idea of developing a “behavioral signature” – asking for behavior instead of self-reported evaluations and exploiting the idea of opportunity costs – is really nice, my main concern, as also outlined in the point above, is that I was left questioning the validation of the measure and the added value.

This comment converges with Comment 2 by Reviewer 1. We reproduce our responses to those comments here for this reviewer’s convenience.

Response to C2.R1: *“A subset of our post-hoc follow-up analyses in response to reviewer comments tested whether social value scores were still predictive of social behavior over and above relationship quality and social loss aversion, Pgs. 15-16:*

“Are Social Value Scores Predictive of Behavior When Controlling for Other Facets of Relationships? Having further established the reliability and validity of social value scores, we next checked whether social value scores are predictive of behaviors when controlling for other facets of relationships. Given our available data, this meant regressing social choice preferences on the dictator game or the forced choice question about spending time with one of two social partners onto social value scores, relationship quality, and social loss aversion. Because each of the two dependent values requires participants to decide between a pair of known others (e.g., parent versus friend, friend versus acquaintance), we differenced the relationship facets and social value scores before entering them into the model to for better interpretability. To illustrate this, we ran one model in which we subtract relationship facets and social value scores for friend from parent and then entered the differences into a model predicting allocations between parent and friend on the dictator game. We enacted this procedure in the three aforementioned samples using Bayesian regression (all predictors were standardized and received standard normal priors).

Results from the models are depicted in Supplementary Figures 4 – 5. Overall, social value scores showed modest but consistent associations with outcomes and most of the posterior mass in these analyses usually fell around either side of zero. Some associations between social value and choice behavior controlling for other relationship facets were stronger, but these results were inconsistent. These analyses suggest that social value is a modest yet persistent predictor of social behavior after adjusting for other facets of social relationships.”

The reviewer can thus see that social value scores are thus less predictive of the behavioral measures in question than other facets of relationships. Crucially, in light of the additional post-hoc analyses further validating the signature, we view this as an interesting substantive finding, rather than something that fully invalidates our proposed method for engineering a behavioral signature of social value. We have amended the discussion section to that effect, Pgs. 16-17:

“Indeed, our results suggest that interpersonal relationship value is related to, but ultimately distinct from, relationship quality, as well as several other ways of evaluating relationships. While our findings also show that social value is a motivator of social behavior, our post hoc analyses lead us to conclude that other ways of assessing relationships (social loss aversion, relationship quality) are stronger predictors of the behavioral measures used here. Given that our method for calculating social value appears to yield reliable and valid scores, we interpret this finding as helping delineate the facets of social relationships that likely shape social behavior. Our evidence suggests that social value has a small effect in shaping behavior, but is partially eclipsed by relationship quality and social loss aversion. These findings are non-trivial in our view because they help untangle the nomological network of associations between relationship facets and social behavior.”

To reiterate a sentiment expressed above: we agree with the reviewer that it would be interesting if social value scores outperformed other facets of relationships in predicting social behavior. Yet, since we present solid evidence on the reliability and validity of the proposed method, we argue that it is *also* interesting that social value scores show an (albeit smaller) association with social behavior after controlling for relationship quality and social loss aversion. This is a

meaningful substantive finding in its own right because we are teasing apart how related but distinct facets of relationships are differentially linked to social behavior.

(6) Information of the measures in the main-text

The writing of the paper was sometimes hard to follow. I understand that the structure of the journal requires the methods to be after the main text. Yet, the authors chose to provide a lot of statistical details also in the main text, while offering little information on the actual measurements and elicitation methods. It would really help to move some of the information on the measurements (what was exactly measured and how?) to the main text, as this is such a central part of the paper and is difficult to understand without consulting the methods (and SI), first.

We agree with the reviewer that the structure of the paper could be rearranged to improve clarity. To do this, we have moved the methods section to appear before the results section (in response to Comment 3 by Reviewer 1) and now we added information about the activity sourcing and likelihood rating prompts to the main text. We are happy to move additional materials from the Supplement to the main text if the reviewer feels it will be helpful to readers.

(7) The distinction between friend and acquaintance already seems to imply a value judgment of different relationships. In that sense, it is a bit tautological. But this could actually be exploited more: I apologize if I missed that, but do people have lower likelihood ratings overall when paired with an acquaintance vs. a friend, and how large is this effect? This could serve as a simple and straightforward ‘manipulation check,’ increasing trust in the validity of this component of the measurement.

The reviewer raises a good point. We had previously included paired differences in social value scores between friends and acquaintances, but had not done so on the ‘raw’, unit-weighted likelihood scores. We computed such scores in two confirmatory samples – prolific data with SONA-sourced activities, and prolific data with MTurk-sourced activities. We used these two datasets to be consistent with the *post hoc* analyses in the revised version of the manuscript.

We observed a pattern of differences between friend and acquaintance unit-weighted likelihood ratings in line with the reviewer’s intuition: participants were overwhelmingly likely to endorse wishing to spend their leisure time with their friend relative to the acquaintance (results displayed below).

We are currently omitting these results from the manuscript out of caution to not oversaturate what is already a dense manuscript, but we are happy to add these findings in if the reviewer feels it would strengthen the manuscript.

Confirmatory Phase Data, SONA-sourced activities

Cohen’s $d_{\text{friend} - \text{acquaintance}}$: 0.960

Mean(SD) friend unit-weighted likelihood rating: 4.130(1.264)

Mean(SD) acquaintance unit-weighted likelihood rating: 3.128(1.384)

Confirmatory Phase Data, MTurk-sourced activities

Cohen's $d_{\text{friend} - \text{acquaintance}}$: 1.272

Mean(SD) friend unit-weighted likelihood rating: 4.152(1.130)

Mean(SD) acquaintance unit-weighted likelihood rating: 2.843(1.298)

Dear Dr Guassi Moreira,

Thank you for your patience during the peer-review process. Your manuscript titled "A Behavioral Signature for Quantifying the Social Value of Interpersonal Relationships with Specific Others" has now been seen by 3 reviewers, and I include their comments at the end of this message. They find your work of interest but raised some important points. We are interested in the possibility of publishing your study in *Communications Psychology*, but would like to consider your responses to these concerns and assess a revised manuscript before we make a final decision on publication.

We therefore invite you to revise and resubmit your manuscript, along with a point-by-point response to the reviewers. Please highlight all changes in the manuscript text file.

Editorially, we consider particularly important that you address the measurement invariance concern raised by R1. Your revision should also take into account the other comments and suggestions of improvements raised by all reviewers.

I am attaching an Editorial Requests Table that details critical reporting requirements for the revised manuscript. Please attend to each item and ensure your manuscript is fully compliant. We are requesting that your manuscript aligns with these requirements as this facilitates the evaluation of your manuscript, reducing delays in re-review and potential future acceptance. If your revised manuscript is not aligned with these requests on major issues, such as those concerning statistics, it may be returned to you for further revisions without re-review. Additional information can be found in our style and formatting guide *Communications Psychology* formatting guide.

Please use the following link to submit your

- revised manuscript,
- point-by-point response to the referees' comments,
- cover letter (as a separate document),
- the Editorial Policy Checklist (see below),
- the Reporting Summary (see below), and
- the completed Editorial Request Table (attached):

[link redacted]

Best regards,

Mael Lebreton

Mael Lebreton, PhD

Editorial Board Member

Communications Psychology

orcid.org/0000-0002-2071-4890

REVIEWERS' EXPERTISE:

Reviewer #1: valuation, social decision making

Reviewer #2: valuation, social decision making

Reviewer #3: groups, social decision making

REVIEWER REPORTS:

Reviewer #1 (Remarks to the Author):

I appreciate the authors' thorough revisions. I believe most of my comments have been addressed, and the manuscript is much improved. Nevertheless, there are a few remaining issues that warrant some (minor) revisions.

1) Measurement invariance. Although the authors add metrics and justification for unidimensionality, they did not address the point about measurement invariance. Please add appropriate statistics. If measurement invariance is weak, then I think it's worth noting this limitation in the discussion as something for future work to build on.

2) Data in Supp Figures 3 and 4. It's clear that the difference in social value was not as strong of a predictor as other metrics, but the graphs are not compelling and I suspect the text needs more details since all of the confidence intervals appear to overlap with zero.

3) Feasibility. Can the authors please add concrete details about how long this measure actually took rather than estimating 5 minutes? Are these data not readily available?

Reviewer #2 (Remarks to the Author):

The authors have adequately addressed my previous comments.

Reviewer #3 (Remarks to the Author):

I very much appreciate the elaborate responses of the authors and the extensive revisions in the manuscript. The presentation of the method details in the manuscript makes it much easier to

follow the logic and rationale and the section “Further Psychometric Assessment of Social Value Scores” addresses many of the concerns raised before.

While I would judge the work to be a first (valuable) step into creating a “behavioral signature for quantifying the social value” (as more could be done to test reliability and validity), I think the work can motivate future investigations in this direction and merits publication.

As such, I only have some comments left that should be easy to address;

(1) “Interrater reliability – How idiosyncratic are [...]” preferences? (MaxDiff approach; point raised before)

I understand the dilemma here (random ordering and different subsets across participants). The approach of scaling seems to make sense. However, I am still struggling with the consequence of this scaling approach that the authors employ;

Let’s assume there are some time-spending preferences that are very consistent across participants (e.g., everybody agrees to prefer to spend time for “going out to eat” compared to other activities) whereas other preferences are highly idiosyncratic (like some people enjoy “visiting a gun range for recreational shooting”, others do not like to spend time on this at all and some are somewhat in the middle).

When constructing questionnaires, this could be akin to items that have clear factor loadings and others that are just very noisy; from my understanding, in the construction of questionnaires, one would just drop items that are too noisy in the construction phase and leave the items that have good quality.

My question is; The scaling seems to change the rank-order of the item (first to third column in Table S2/S3), but wouldn’t it be more straightforward to just remove activities for which the underlying time-spending preferences across participants was too idiosyncratic and select items that are consistently ranked (i.e., that have low ‘activity variance’)?

This point and the problem of “preference consistency” in the MaxDiff approach should be at least discussed and could be addressed in future work.

(2) Framing of the questions and underlying assumptions

I am still wondering about the framing of the questions; “Which of the following activities are you most likely to do in this time?” I am sorry to repeat this point but, since the authors refer to value-based decision making, the authors should discuss the problem of (hidden) costs that people may factor into their decision in the MaxDiff task. For example, listening to music is “easy” but maybe not as valuable, whereas “spending time at an amusement park” is more “difficult” (travel time, financial costs, etc.) but maybe seen as more valuable. Hence, people may say that they are most likely to “listen to music” when they “had a couple hours of time wherein [they] had no obligations or commitments” but, if they could, would rather spend time at an amusement park (just to make an arbitrary example from the item set). From my reading, the danger is that the rankings therefore do not reflect ‘pure’ value (keeping costs constant) but a combination of subjective value and costs. The point is that frequency (“most likely to do”) is not the same as how much something may be “valued”. Yet, that is what the authors seem to suggest (e.g., “behavioral signature of social value” – caption of Fig.1). The weights in S2/S3 seem to corroborate that; Items with high weight seem to be items that people simply do often, because they are easy to do (e.g., browsing the internet, conversating with someone) whereas items that may be highly enjoyable (but uncommon) are ranked lower (e.g., attending a sporting event, hosting a social gathering). That choices reveal preferences usually assumes that costs are kept constant or that costs are explicitly integrated in the decision and taken into account. This could be at least discussed.

EDITORIAL POLICIES

We ask that you ensure your manuscript complies with our editorial policies and reporting requirements.

To that end, we require revised manuscripts to be accompanied by two completed items: a reporting summary that collects information on study design and procedure, and an editorial policy checklist that verifies compliance with all required editorial policies.

Nature Research Reporting Summary

Editorial Policy Checklist

All points on the policy checklist must be addressed. Your revised manuscript can only be sent back to the referees if these checklists are completed and uploaded with the revision.

Notes: If you have submitted a Stage 1 Registered Report, Review, Primer, Comment, or Perspective you do not need to submit these forms. If you have already submitted these forms, you may disregard this request.

* TRANSPARENT PEER REVIEW: Communications Psychology uses a transparent peer review system. This means that we publish the editorial decision letters including Reviewers' comments to the authors and the author rebuttal letters online as a supplementary peer review file. However, on author request, confidential information and data can be removed from the published reviewer reports and rebuttal letters prior to publication. If your manuscript has been previously reviewed at another journal, those Reviewers' comments would not form part of the published peer review file.

Author Responses: second round.

Revision of COMMSPSYCHOL-24-0016

A Behavioral Signature for Quantifying the Social Value of Interpersonal Relationships with Specific Others

General Comments

We appreciate the thoughtful reviewer feedback. In reading over their comments, it is clear they took a lot of time and effort to provide constructive feedback to help better the manuscript. We are deeply appreciative of the work they put into their reviews and have made every effort of our own to thoroughly address their concerns and incorporate their suggestions.

We comprehensively addressed all reviewer comments. For example, the manuscript now contains more substantive consideration of future directions, we have expanded our discussion of the behavioral signature with respect to effort cost and individual differences, and we now report the results of additional psychometric analyses. We have also completed the Editorial Policy Checklist (entered through the submission portal).

As we detail in our response responses below, we amended the text of the manuscript where necessary (e.g., inserting additional considerations in the discussion) and conducted thorough follow-up statistical analyses to address lingering concerns about the reliability and validity of our method for calculating social value. Our detailed, point-by-point responses to each reviewer's concerns are included below.

Overall, we believe our responses to the comments raised by the editor and reviewers have greatly strengthened the paper and we are confident that it is now more suitable for publication in *Communications Psychology*.

Reviewer 1

1. Measurement invariance. Although the authors add metrics and justification for unidimensionality, they did not address the point about measurement invariance. Please add appropriate statistics. If measurement invariance is weak, then I think it's worth noting this limitation in the discussion as something for future work to build on.

We apologize for not addressing this issue in the previous version of the manuscript. The edits in the last revision were quite extensive and we inadvertently overlooked this particular matter. To address the matter now, we ran a series of latent variable models to test for measurement invariance. We focused on testing for configural invariance, which establishes equivalence in the factor structure, because other forms of measurement invariance involve factor loadings and intercepts and these quantities are not relevant here because we are not unit-weighting each individual item. We opted to test for a unidimensional factor structure because we previously found the factor structure of the likelihood ratings to be essentially unidimensional (i.e., most of

the observed variance was accounted for by a general factor common to all items). Any given model constrained the factor structure to be the same between the two repeated measurements.

To make inferences about measurement invariance, chose to interpret 2 fit statistics: Root Mean Squared Error of Approximation (RMSEA) and the Comparative Fit Index (CFI). The former is a metric of absolute fit, whereas the latter is a metric of relative fit (compared to a saturated and model). We performed measurement invariance analyses on the confirmatory subsamples obtained from prolific (one subsample was administered using SONA-sourced weights; the other administered using MTurk-sourced weights). For each subsample, we fit three models, each testing equivalence between all pairwise comparisons of social partners (parent and friend; parent and acquaintance; friend and acquaintance). We first attempted to test equivalence in one modeling step but this model did not converge and so opted to proceed with pairwise tests. Modeling code is now included in the study’s repository on OSF. Results are provided in the tables below, broken down by subsample (SONA-sourced weights or MTurk-sourced weights).

SONA-sourced weights

	Parent - Friend	Parent - Acquaintance	Friend - Acquaintance
RMSEA	0.077	0.077	0.079
CFI	0.499	0.518	0.507

MTurk-sourced weights

	Parent - Friend	Parent - Acquaintance	Friend - Acquaintance
RMSEA	0.082	0.079	0.082
CFI	0.428	0.506	0.476

The reviewer will likely immediately note the discrepancy between the acceptably fitting RMSEA values and the poorly fitting CFI values. That is, RMSEA values indicate an acceptable, if not ideal, degree of measurement invariance whereas the CFI values suggest otherwise. We think this discrepancy arises due to the difference in fit indices (absolute vs relative) and the nature of the likelihood ratings’ factor structure. Because CFI compares model fits relative to a ‘best’ fitting model (i.e., saturated model), this metric is likely more sensitive to differences in factor model specification. RMSEA, on the other hand, quantifies the degree of absolute misfit and is less susceptible to differences in the exact factor structure of the model because it is not being compared to a baseline model. While it would be ideal to have both indices show excellent results, we are comfortable concluding an acceptable degree of measurement invariance since the absolute fit of the configural invariance models is high. Moreover, we note again the *exact* nature of the factor structure is irrelevant for our purposes – we previously just needed confirmation that the scores could be deployed as a single composite (i.e., the factor structure needed to be

essentially or practically unidimensional), and we previously observed evidence to indicate this is the case. A reliance on the CFI is thus misleading or less germane to the validity of the measure—nevertheless, we report the results here in the interest of transparency.

We intended to test for measurement invariance by online population and age, but when subsetting the data for analysis, there were not enough individuals (more than 70 or 56, given the activity signature) in a given group to suitably perform the analysis. More specifically, when looking to test differences in population (MTurk vs SONA participants), we were forced to use exploratory subdatasets. However, none of the available datasets had the requisite sample size for both groups.

We attempted to at least approximate differences in age by binning age from the confirmatory, prolific dataset in college (18-22) and non-college age (23-30), but this also did not result in enough participants in the college age group to perform analyses.

While it is disappointing that we could not test this, we do note that the MTurk signature was developed on a wider age range than the SONA signature, but results from both signatures still give comparable results. This does not obviate the need for future testing, but is at least modestly reassuring at present.

We now report this information in the manuscript (pg. 14): “As part of this process, we also tested whether the activity weights evinced measurement invariance between social partners. We focused on testing for configural invariance, which establishes equivalence in the factor structure, because other forms of measurement invariance involve factor loadings and intercepts and these quantities are not relevant here because we are not unit-weighting each individual item. We found moderate evidence for configural invariance, as indexed by RMSEA (root mean squared error of approximate) absolute fit statistics (all values between .07 and .08 in all confirmatory subsamples). We could not test for invariance based on age (college age vs non-college age) or online sample (SONA vs MTurk) due to sample size constraints.”

2. Data in Supp Figures 3 and 4. It's clear that the difference in social value was not as strong of a predictor as other metrics, but the graphs are not compelling and I suspect the text needs more details since the all of the confidence intervals appear to overlap with zero.

We regret that we were not sufficiently clear in the figures and have worked to clarify the figure captions. The relevant figure captions (Figs. 2-4; Supplementary Figs. 3-4) now note that the error bars correspond to 89% highest density credible intervals, which have different inferential criteria than confidence intervals. The captions of those figures now provide more details and encourage readers to visit our section in the manuscript on inferential criteria to assist their interpretation of the plots. Specifically, the captions for these figures now state:

“Evidence was judged to be robust if the HDI did not include 0 or the HDI fell outside of the Region of Practical Equivalence (ROPE) and moderate if part of the HDI fell outside of ROPE (see “Inferential Criteria” section of the main text). ROPE was defined as the range between -0.1 to 0.1.”

We initially also intended to depict ROPE in relevant plots (i.e., plots where whether or not the HDI partially falls outside of ROPE informs the evaluation of evidence strength – specifically, Supplementary Figures 3-4 and Figure 3’s bottom right panel) as part of this revision. However, the high variability of posterior samples across analyses complicated efficient visual depiction. We can implement this edit at the reviewer’s request, but have tentatively avoided doing so to avoid adding visual clutter. Instead, we have added brief statements summarizing the strength of the evidence (using the criteria summarized in the “Inferential Criteria” section of the paper, which is now reiterated in these figures’ captions, as noted above) for the relationships between social value scores and other variables to Figures 2-4 and Supplementary Figures 3-4. The revised captions for these figures now contain the following text:

Figure 2: “Here, there was robust evidence for an association between social value scores and relationship quality for all three social partners. There was also robust evidence for an association between social value scores and social loss aversion for all three social partners.”

Figure 3: “Here, there was robust evidence that social value scores predict choice preferences involving monetary outcomes for all social partners. There was also robust evidence that social value scores predict choice preferences involving social outcomes when pitting friends or acquaintances against parents. There was moderate evidence that social value scores predict choice preferences involving social outcomes when pitting friends against acquaintances.”

Figure 4: “Here, there was robust evidence that social value scores predict choices for all social partners for both monetary and social outcomes in a multi-trial social decision paradigm.”

Supplementary Figure 3: “Here, when controlling for relationship quality and social loss aversion, there was robust evidence that social value scores were associated with choice preferences regarding whether participants would rather spend time with their parent or friend and moderate evidence that social value scores were associated with choice preferences regarding whether participants would rather spend time with their parent or acquaintance and whether they would rather spend time with their friend or acquaintance. Comparable evidence was not found for choices in the dictator game.”

Supplementary Figure 4: “Here, when controlling for relationship quality and social loss aversion, there was moderate evidence that social value scores were associated with choice preferences regarding whether participants would rather spend time with their parent or friend, whether they would rather spend time with their parent or acquaintance, and whether they would rather spend time with their friend or acquaintance. Comparable evidence was not found for choices in the dictator game.”

We are also open to further suggestions to amending the text in the main document regarding the results depicted in Supplementary Figures 3-4, which reads: “Overall, social value scores showed modest but consistent associations with outcomes. While we generally found moderate evidence for these associations (i.e., part of each HDI fell outside of ROPE; see “Inferential Criteria”), most of the posterior mass in many of these analyses fell around either side of zero (i.e., overlapped with zero). Some associations between social value and choice behavior controlling for other relationship facets were stronger, but these results were inconsistent.”

We believe this text adequately communicates what is depicted in the plots – moderate evidence for small associations with wide credible intervals, save for a few cases.

We also now distinguish between confidence and credible intervals (pg. 8): “While credible intervals are often interpreted similarly to confidence intervals, we note that a credible interval overlapping with zero is not necessarily indicative of a null statistical effect. Credible intervals simply help summarize the distribution of evidence over parameter space, and overlap with zero does not rule out evidence in favor of a meaningful effect.”

3. Feasibility. Can the authors please add concrete details about how long this measure actually took rather than estimating 5 minutes? Are these data not readily available?

Thank you for raising this point. We have some data that can speak to this, which we now mention in the main text, as described in more detail below.

We implemented all surveys through the Qualtrics platform. By default, Qualtrics logs the total time it takes to complete the survey for every participant. While it can also log the time it takes for each participant to complete a block of questions, unfortunately, we did not have the foresight to isolate the behavioral signature questions to a single block and then program the survey to record completion times for said block. Qualtrics used to have a free tool that could be used to estimate the time to complete the behavioral signature for one individual social partner, but this tool is unfortunately no longer available to us (it is either priced out of our institution’s subscription or Qualtrics removed the functionality entirely; their documentation is not entirely clear to us).

We agree with the reviewer that practical information about implementation time is nevertheless important to provide. To give readers *some* data-informed estimate, we enacted the following procedure. First, we selected a sub-study dataset with the shortest survey. In other words, this survey has the smallest number of additional items besides the behavioral signature likelihood ratings (demographics, IPPA for each individual social partner, one-shot questions for each individual social partner, and ‘head-to-head’, one-shot social decision items with all pairwise social decision partner combinations). In total, this survey had 339 items and it was collected on a sample of thirty undergraduate psychology students using the SONA-sourced activities (70 activities). We divided the number of behavioral signature likelihood rating questions (70 activities * 3 social partners = 210 questions) by the total number of survey items (339). We then multiplied this value by each participant’s response time for the entire survey (in minutes) to estimate how long it would take to complete all three sets of likelihood ratings. This value was further divided by three to isolate the time it would take to complete for a single social partner. Results are listed below:

Time to complete likelihood ratings for all 3 social partners

Mean: 20.24 minutes

SD: 23.93 minutes

Range: [8.94, 138.88]

Time to complete likelihood ratings for one social partner

Mean: 6.75 minutes

SD: 7.98 minutes

Range: [2.98, 46.29]

(The reviewer can probably surmise the presence of outliers from these summary statistics. Indeed, there were two extreme cases that took a very long time to complete the entire survey. The 3 partner and single partner means drop to 14.82 and 4.94 minutes when removing these outliers. There were no outliers in the other direction, i.e., who completed the survey too quickly.)

We have edited the manuscript to be more descriptive, pg. 18: “Last, administering the likelihood ratings can be achieved quickly: We analyzed survey duration data and estimate that it typically takes approximately 4.94 to 6.75 minutes to complete said ratings for one target individual.”

We are happy to repeat this procedure for all sub-study datasets and/or include more detailed information about these calculations in the main document if the reviewer feels it would strengthen the submission. We chose the shortest survey under the logic that administration in a short survey would avoid any potential additive or interactive effects with participant fatigue that may set in with larger surveys. In other words, this best approximates the ‘pure duration’ of the likelihood ratings if one were to administer them in a survey with no other items.

Reviewer 3

1. “Interrater reliability – How idiosyncratic are [...]” preferences? (MaxDiff approach; point raised before)

I understand the dilemma here (random ordering and different subsets across participants). The approach of scaling seems to make sense. However, I am still struggling with the consequence of this scaling approach that the authors employ;

Let’s assume there are some time-spending preferences that are very consistent across participants (e.g., everybody agrees to prefer to spend time for “going out to eat” compared to other activities) whereas other preferences are highly idiosyncratic (like some people enjoy “visiting a gun range for recreational shooting”, others do not like to spend time on this at all and some are somewhat in the middle).

When constructing questionnaires, this could be akin to items that have clear factor loadings and others that are just very noisy; from my understanding, in the construction of questionnaires, one would just drop items that are too noisy in the construction phase and leave the items that have good quality.

My question is; The scaling seems to change the rank-order of the item (first to third column in Table S2/S3), but wouldn't it be more straightforward to just remove activities for which the underlying time-spending preferences across participants was too idiosyncratic and select items that are consistently ranked (i.e., that have low 'activity variance')?

This point and the problem of "preference consistency" in the MaxDiff approach should be at least discussed and could be addressed in future work.

We appreciate the reviewer's thoughtful comments on the psychometric aspects of the MaxDiff component of engineering the behavioral signature. We have addressed this concern in two key ways.

We first tackled the reviewer's latest concern about high variance activities. We did this by correlating social value scores computed using our original method with scores computed using an alternative method that minimized the influence of high variance weights, as suggested by the reviewer. Our logic here is that since the reviewer's concern is partly about rank order changes in activity weights, we extrapolate that they are concerned that such changes would also change the rank ordering of participant social value scores. Checking the correlation of social value scores computed using these two methods provides an efficient way of verifying this.

'Removing activities that are too idiosyncratic' (to paraphrase) could be interpreted a few different ways. To address the matter comprehensively, we re-computed social value scores by (i) varying the activity weight scaling (2 options: use scaled weights or raw, unscaled weights) and (ii) masking out (i.e., removing) high variance activities at two different thresholds (2 options: removing activities with weights in the top 10% and 25% in terms of their random effect variance). Fully crossing these two specifications yielded 4 total alternate ways of computing social value scores.

We performed these analyses on two confirmatory subsamples collected from Prolific, one that was administered the signature with SONA-sourced activities and the other administered MTurk-sourced activities. We then correlated each of these variants with the normally computed social value scores for each social partner (parent, friend, acquaintance). As a reminder, our normal method for computing social value scores is to take the dot product between the scaled activity weights (activity weight / random effect variance of activity weight).

Removing the activities with the top 10% and 25% most variable weights from the original 70-activity SONA-sourced signature corresponded to retaining 63 and 52 activities, respectively. Doing the same for the original 56-activity MTurk-sourced signature corresponded to retaining 49 and 42 activities, respectively.

Results indicated very high correlations between the social value scores calculated using these approaches, as indicated in the tables below, broken down by social partner:

Parent social value scores – Correlation between social value scores as normally calculated (all weights, scaled) and using alternative approaches that vary scaling of weights and inclusion of activities with high-variance weights

Alternative Approach for Calculating Social Value Scores	SONA	MTurk
Scaled Weights – Mask Out Top 10% High Variance Items	0.997	0.987
Scaled Weights – Mask Out Top 25% High Variance Items	0.983	0.979
Raw Weights – Mask Out Top 10% High Variance Items	0.963	0.871
Raw Weights – Mask Out Top 25% High Variance Items	0.930	0.862

Friend social value scores – Correlation between social value scores as normally calculated (all weights, scaled) and using alternative approaches that vary scaling of weights and inclusion of activities with high-variance weights

Alternative Approach for Calculating Social Value Scores	SONA	MTurk
Scaled Weights – Mask Out Top 10% High Variance Items	0.997	0.981
Scaled Weights – Mask Out Top 25% High Variance Items	0.985	0.972
Raw Weights – Mask Out Top 10% High Variance Items	0.965	0.812
Raw Weights – Mask Out Top 25% High Variance Items	0.938	0.815

Acquaintance social value scores – Correlation between social value scores as normally calculated (all weights, scaled) and using alternative approaches that vary scaling of weights and inclusion of activities with high-variance weights

Alternative Approach for Calculating Social Value Scores	SONA	MTurk
Scaled Weights – Mask Out Top 10% High Variance Items	0.997	0.987
Scaled Weights – Mask Out Top 25% High Variance Items	0.989	0.981
Raw Weights – Mask Out Top 10% High Variance Items	0.953	0.835

Raw Weights – Mask Out Top 25% High Variance Items	0.932	0.824
--	-------	-------

The supplement of the manuscript, where the random effect variance scaling is described, has been updated to reflect the knowledge gained from these sensitivity analyses (pg. 32): “Various sensitivity analyses revealed that this scaling and inclusion of activities with high random effect variance did not appreciably change the rank order among subjects for any of the social targets tested here. More specifically, correlations between social value scores computed using our original approach (with all activities included and with scaled weights) and social value scores computed without scaling weights by their variance and/or by excluding activities with high-variance weights ranged from $r = 0.81$ to 0.98 across social partners.”

Second, we address the reviewer’s previous issue of preference consistency in the MaxDiff data. Previously we noted that calculating inter-rater agreement was difficult because of “the random ordering and subsetting of the MaxDiff design” (response to Reviewer 3 – Comment 2 from the prior round of reviews). While we still believe this information to be relevant and important to keep in mind throughout the rest of this section, it since occurred to us that there is another potential way to gauge inter-rater agreement/consistency. We would like to preface our description of this method and its results by noting that this method is imperfect, given what we noted above about the MaxDiff design, in which activities are not only randomly ordered but also randomly subsetted across participants; thus, the results of the analysis described below should be interpreted with caution.

We calculated subject-specific activity weights from both MaxDiff models by taking the posterior mean of each subject-specific random effect and adding it to the posterior mean of the ‘fixed effect’ activity weight estimates. This yields a ‘rough’ estimate of each participant’s unique preferences among the activities. We emphasize ‘rough’ here because of the factors above (different set composition across participants due to the random subsetting). We computed the mean pairwise correlation (Pearson’s r) across these participant-specific preferences to arrive at an estimate of inter-rater reliability.

Both the SONA-sourced and MTurk-sourced activity weight sets evinced moderate inter-rater reliability – SONA: 0.486, MTurk: 0.505. According to several benchmarks (Landis & Koch, 1977; Fleiss, 1981; Altman, 1991; Koo & Li, 2016), these values fall in ranges that are labeled with descriptors such as ‘moderate’, ‘intermediate’, and ‘acceptable’. Given the randomization of the MaxDiff design should, in theory, only introduce noise, we believe these values reflect a lower bound for the activity weights’ reliability. We are happy to add these results to the manuscript if the reviewer feels it would strengthen the submission.

Collectively, we think that these two pieces of evidence help assuage concerns about the reliability of the activity weights used to calculate social value scores.

We also agree that it is important to discuss the issue of preference consistency in participants' activity ratings as measured using the MaxDiff approach and how it could be addressed in future work. As such, we have added the following text to the revised Discussion section (pg. 19 of the revised manuscript):

“Of these potential extensions, we see two as particularly important. The first involves the use of personalized activity weights. Although the inclusion of high-variance weights in the behavioral signature did not make a difference in terms of the rank ordering of participants' social value scores (see Supplement), future versions of this approach to calculating social value would likely benefit from additional precision if personalized weights were incorporated in some way. Doing so would mitigate any noise that is contributed by inconsistency across participants in their activity preferences. If it is not feasible to complete the weight-generating procedure for each individual subject, then researchers could adopt a method where a single pre-computed weight set is used (as we do here), and low-variance activity weights are retained from the overall model estimate but high-variance activity weights are replaced with subject-specific weights from the pre-computed participants by matching individuals from the pre-computed sample and the researcher's focal sample on relevant demographic (e.g., age, sex) or psychological (e.g., personality traits) dimensions by using propensity score matching or another approach.”

2. Framing of the questions and underlying assumptions

I am still wondering about the framing of the questions; “Which of the following activities are you most likely to do in this time?” I am sorry to repeat this point but, since the authors refer to value-based decision making, the authors should discuss the problem of (hidden) costs that people may factor into their decision in the MaxDiff task. For example, listening to music is “easy” but maybe not as valuable, whereas “spending time at an amusement park” is more “difficult” (travel time, financial costs, etc.) but maybe seen as more valuable. Hence, people may say that they are most likely to “listen to music” when they “had a couple hours of time wherein [they] had no obligations or commitments” but, if they could, would rather spend time at an amusement park (just to make an arbitrary example from the item set). From my reading, the danger is that the rankings therefore do not reflect ‘pure’ value (keeping costs constant) but a combination of subjective value and costs. The point is that frequency (“most likely to do”) is not the same as how much something may be “valued”. Yet, that is what the authors seem to suggest (e.g., “behavioral signature of social value” – caption of Fig.1). The weights in S2/S3 seem to corroborate that; Items with high weight seem to be items that people simply do often, because they are easy to do (e.g., browsing the internet, conversating with someone) whereas items that may be highly enjoyable (but uncommon) are ranked lower (e.g., attending a sporting event, hosting a social gathering). That choices reveal preferences usually assumes that costs are kept constant or that costs are explicitly integrated in the decision and taken into account. This could be at least discussed.

We agree with the reviewer and devote more space in the discussion for considering this issue (pg. 19):

“The second significant potential extension involves better incorporating the effort of each activity that comprises the behavioral signature. Addressing this current limitation is important because inadvertent conflation with effort may dilute or bias the calculation of social value. This could be addressed by modifying future prompts when collecting data for calculating activity weights or likelihood ratings, or obtaining independent ratings of effort for each activity and factoring them into the calculation.”

Decision letter and referee reports: third round

Dear Dr Guassi Moreira,

Your manuscript titled "A Behavioral Signature for Quantifying the Social Value of Interpersonal Relationships with Specific Others" has now been seen by our reviewer, whose comments appear below. In light of their advice I am delighted to say that we are happy, in principle, to publish a suitably revised version in *Communications Psychology*.

We therefore invite you to revise your paper one last time to address the remaining concerns of our reviewers and a list of editorial requests. At the same time we ask that you edit your manuscript to comply with our format requirements and to maximise the accessibility and therefore the impact of your work.

EDITORIAL REQUESTS:

SUBMISSION INFORMATION:

In order to accept your paper, we require the files listed at the end of the Editorial Requests Table; the list of required files is also available at <https://www.nature.com/documents/commsj-file-checklist.pdf> .

OPEN ACCESS:

* **DATA AVAILABILITY:**

(link redacted)

Best regards,

Jennifer Bellingtier

Jennifer Bellingtier, PhD
Senior Editor
Communications Psychology

Mael Lebreton, PhD
Editorial Board Member
Communications Psychology
orcid.org/0000-0002-2071-4890

REVIEWERS' EXPERTISE:

Reviewer #1: valuation, social decision making

REVIEWERS' COMMENTS:

Reviewer #1 (Remarks to the Author):

I appreciate the authors' thorough revisions. My comments have been adequately addressed, and I do not need to see manuscript again.

Nevertheless, I do encourage the authors to look into the measurement invariance issue a little more. Non-convergence at the configural level are notable, especially given that the model is not complicated (i.e. unidimensional). One potential cause for non-convergence at configural level could be that the unidimensional factor structure they used did not hold across all three social groups (i.e. when parent, friend, acquaintance are analyzed together). I think this limitation (and the associated statistics) would be worth briefly mentioning in the supplement and potentially the discussion section.

Author Responses: third round.

Revision of COMMSPSYCHOL-24-0016

A Behavioral Signature for Quantifying the Social Value of Interpersonal Relationships with Specific Others

General Comments

We appreciate the additional feedback provided by Reviewer 1 about clarifying the consequences of potential measurement invariance. We have added additional language to manuscript on this matter below.

We are sincerely grateful for the editor and reviewers' time and effort during the peer review process. We are excited that our manuscript is now ready for publication in *Communications Psychology*.

Reviewer 1

1. I appreciate the authors' thorough revisions. My comments have been adequately addressed, and I do not need to see manuscript again.

Nevertheless, I do encourage the authors to look into the measurement invariance issue a little more. Non-convergence at the configural level are notable, especially given that the model is not complicated (i.e. unidimensional). One potential cause for non-convergence at configural level could be that the unidimensional factor structure they used did not hold across all three social groups (i.e. when parent, friend, acquaintance are analyzed together). I think this limitation (and the associated statistics) would be worth briefly mentioning in the supplement and potentially the discussion section.

We have revised the post-hoc psychometric analysis section of the manuscript to better identify this to readers, briefly explain why it might be happening, and flag it as an area of concern for future research (pg. 14): “As part of this process, we also tested whether the activity weights evinced measurement invariance between social partners. We focused on testing for configural invariance, which establishes equivalence in the factor structure, because other forms of measurement invariance involve factor loadings and intercepts and these quantities are not relevant here because we are not unit-weighting each individual item. We found moderate evidence for configural invariance, as indexed by RMSEA (root mean squared error of approximate) absolute fit statistics (all values between .07 and .08 in all confirmatory subsamples). However, using the comparative fit index (CFI) revealed statistics that were very poor (in .42 - .52 range). One explanation for this could be due to the fact that while the factor structure is essentially unidimensional for all three sets of social partners independently (based on the high $\omega_{\text{hierarchical}}$ values), a strict test of this assumption fails. We view this as important to test in future work but remain cautiously optimistic in the interim. We could not test for invariance based on age (college age vs non-college age) or online sample (SONA vs MTurk) due to sample size constraints.”